# Extending the Context of Pretrained LLMs by Dropping Their Positional Embeddings

**Yoav Gelberg**[*]
University of Oxford
yoav@robots.ox.ac.uk

**Koshi Eguchi**
Sakana AI

**Takuya Akiba**
Sakana AI

**Edoardo Cetin**
Sakana AI
edo@sakana.ai

## Abstract

So far, expensive finetuning beyond the pretraining sequence length has been a prerequisite to effectively extend the context of language models (LM). In this work, we break this key bottleneck by ***Dro**pping the **P**ositional **E**mbeddings of LMs after training (DroPE)*. Our simple method is motivated by three key theoretical and empirical observations. First, positional embeddings serve a crucial role during pretraining, providing an important inductive bias that significantly facilitates convergence. Second, over-reliance on this explicit positional information is also precisely what prevents test-time generalization to sequences of unseen length. Third, positional embeddings are not an inherent requirement of effective language modeling and can be safely *removed after pretraining* following a short recalibration phase. Empirically, DroPE yields seamless *zero-shot* context extension *without any long-context finetuning*, quickly adapting pretrained LMs without compromising their capabilities in the original training context. Our findings hold across different models and dataset sizes, far outperforming previous specialized architectures and established rotary positional embedding scaling methods.

## 1 Introduction

Transformers established themselves as the predominant architecture for training *foundation models* at unprecedented scale in language and beyond (Brown et al., 2020; Team et al., 2023; Jumper et al., 2021; Dosovitskiy et al., 2020). The defining feature of transformers is abandoning explicit architectural biases such as convolutions and recurrences in favor of highly general self-attention layers (Vaswani et al., 2017), while injecting positional information about the sequence through positional embeddings (PE) and causal masking. However, despite significant efforts to scale attention to long sequences on modern hardware (Dao et al., 2022;

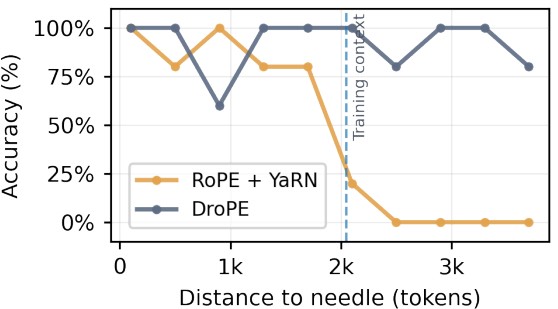

Figure 1: **DroPE generalizes zero-shot to long sequences.** Needle-in-a-haystack retrieval accuracy on sequences at $2\times$ the original context length with *no long context training* (zero-shot context extension).

Liu et al., 2023a; Liu & Abbeel, 2023), this powerful layer is inherently bottlenecked by quadratic token-token interactions, which makes pretraining at long sequence lengths computationally intractable at scale. As a result, enabling models to use contexts beyond their pretraining length *without additional long-context fine-tuning* ("zero-shot context extension") has emerged as a central challenge for the next generation of foundation models (Press et al., 2021; Chi et al., 2023).

When inference sequence lengths exceed the pretraining context, the performance of modern transformer-based LMs degrades sharply. This is directly caused by their use of explicit PEs such as the ubiquitous rotary positional embeddings (RoPE) (Su et al., 2024), which become out-of-distribution at unseen sequence lengths. To address this issue, careful scaling techniques

---

[*]This work was conducted during an internship at Sakana AI.

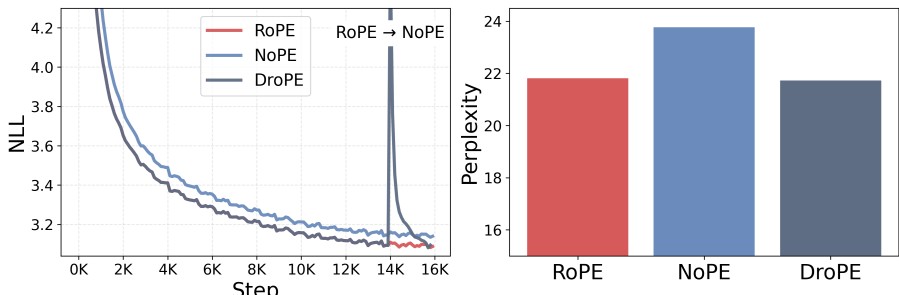

Figure 2: **DroPE matches RoPE's in-context perplexity.** We compare three training recipes: (1) a RoPE transformer trained for 16K steps (16B tokens), (2) a NoPE transformer trained for 16K steps, and (c) a DroPE transformer obtained by training the 14K-step RoPE checkpoint for 2K additional steps. The DroPE recipe matches the RoPE loss within 2K steps and achieves lower final perplexity than the NoPE-from-scratch baseline under the same budget.

that adapt RoPE frequencies on longer sequences were introduced (Chen et al., 2023; bloc97, 2023; Peng et al., 2023; Ding et al., 2024). However, despite their popularity, these methods still rely on an expensive, long-context finetuning phase to *meaningfully* use tokens beyond the original sequence length, failing to generalize out of the box (Lu et al., 2024a). Beyond RoPE transformers, alternative architectures and positional embedding schemes have shown early promise in reducing costs by attenuating the underlying quadratic burden Choromanski et al. (2020); Wang et al. (2020); Xiong et al. (2021); Zaheer et al. (2020) or maintaining better out-of-context generalization (Kazemnejad et al., 2023; Yang et al., 2025b; Puvvada et al., 2025). Yet, these parallel efforts are still far from challenging established pipelines, introducing notable performance and stability trade-offs that prevent wide adoption.

In this work, we challenge the conventional role of RoPE in language modeling, and propose to overcome this inherent trade-off by ***Dro**pping the **P**ositional **E**mbeddings* (DroPE) of LMs after pretraining. Our method is based on three key theoretical and empirical observations. First, explicit positional embeddings significantly facilitate pretraining convergence by baking in an important inductive bias that is difficult to recover from data alone. Second, over-reliance on positional embeddings is precisely what prevents test-time generalization to sequences of unseen length, with RoPE-based context extension methods focusing on recent tokens to retain perplexity. Third, explicit PE is not an inherent requirement for effective language modeling and can be *removed after pretraining*, following a short recalibration phase at the *original context length*.

Empirically, DroPE models generalize zero-shot to sequences far beyond their training context, marking a sharp contrast to traditional positional scaling techniques. Moreover, we show that adapting RoPE models with DroPE does not compromise their original in-context capabilities, preserving both perplexity and downstream task performance. Our findings hold across LMs with different architectures pretrained with up to hundreds of billions of tokens, establishing a new standard for developing robust and scalable long-context transformers.

**Contributions.** In summary, our main contributions are as follows:

(1) In Section 3, we provide empirical and theoretical analysis of the role of positional embeddings in LM *training*, showing their importance in significantly accelerating convergence.

(2) In Section 4, we discuss why RoPE-scaling methods fail to reliably attend across far-away tokens when evaluated zero-shot on long sequences, showing that these approaches inevitably shift attention weights, hindering the model's test-time behavior.

(3) In Section 5, we introduce DroPE, a new method that challenges the conventional role of positional embeddings in transformers, motivated by our empirical and theoretical analyses of its role as a transient but critical training inductive bias.

(4) We demonstrate that DroPE enables *zero-shot generalization* of pretrained RoPE transformers far beyond their original sequence length, *without any long-context finetuning*. DroPE can be incorporated *at no extra cost* into established training pipelines, and can be used to inexpensively empower *arbitrary pretrained LLMs in the wild*.

We **share our full code with this submission** to facilitate future work and extensions toward developing foundation models capable of handling orders-of-magnitude longer contexts.

## 2 PRELIMINARIES

**Self-attention.** Let $h_1^{(l)}, \ldots, h_T^{(l)} \in \mathbb{R}^d$ be the representations fed into the $l$-th multi-head attention block. Queries $q_i$, keys $k_i$, and values $v_i$ are then computed by projecting the inputs $h_i$ via linear layers $W_Q$, $W_K$, and $W_V$. The *attention* operation then computes $T \times T$ matrices of attention scores $s_{ij}$ and weights $\alpha_{ij}$ between all pairs of sequence positions, and reweighs value vectors:

$$s_{ij} = \frac{1}{\sqrt{d_k}} q_i^\top k_j, \quad \alpha_{ij} = \text{softmax}(s_{i1}, \ldots, s_{ii})_j, \quad z_i = \sum_{j \leq i} \alpha_{ij} v_j, \tag{1}$$

where $d_k$ is the head dimension. A multi-head attention block computes multiple attention outputs $z_i^{(1)}, \ldots, z_i^{(H)}$, concatenates them, and projects to the model dimension: $o_i = W_O[z_i^{(1)}, \ldots, z_i^{(H)}]$.

**Language and positional embeddings.** State-of-the-art autoregressive transformer LMs use information about sequence positions provided both implicitly via causal masking of the attention scores[1], and explicitly with positional embeddings (PEs). In particular, the modern literature has settled on the Rotary PE (RoPE) scheme (Su et al., 2024), providing relative positional information to each attention head by rotating $q_i$ and $k_j$ in 2D chunks before the inner product in Equation 1:

$$s_{ij} = \frac{1}{\sqrt{d_k}} (R^i q_i)^\top (R^j k_j) = \frac{1}{\sqrt{d_k}} q_i^\top R^{j-i} k_j, \qquad R = \text{block-diag}\left(R(\omega_1), \ldots, R(\omega_{d_k/2})\right). \tag{2}$$

Here, each $R(\omega_m) \in \mathbb{R}^{2 \times 2}$ is a planar rotation of angle $\omega_m = b^{-2(m-1)/d_k}$ acting on the $(2m, 2m+1)$ subspace of $q_i$ and $k_j$. The base $b$ is commonly taken to be $10^4$.

**Context extension for RoPE.** Given the rapidly growing costs of self-attention, adapting LMs for longer sequences than those seen during training has been a longstanding open problem. To this end, prior context-extension methods introduce targeted rescaling of the RoPE frequencies in Equation 2 to avoid incurring unseen rotations for new sequence positions. Formally, let the training and inference context lengths be $C_{\text{train}} < C_{\text{test}}$, and define the extension factor $s = C_{\text{test}}/C_{\text{train}}$. Context extension methods such as PI (Chen et al., 2023), RoPE-NTK (bloc97, 2023), and the popular YaRN (Peng et al., 2023) define new RoPE frequencies $\omega_m' = \gamma_m \omega_m$ with scaling factors:

$$\gamma_m^{\text{PI}} = \frac{1}{s}, \quad \gamma_m^{\text{NTK}} = \left(\frac{1}{s}\right)^{\frac{2m}{d_k - 2}}, \quad \text{and} \quad \gamma_m^{\text{YaRN}} = (1 - \kappa_m)\frac{1}{s} + \kappa_m, \tag{3}$$

where $\kappa_m \in [0, 1]$ interpolates between 0 and 1 as the base frequency $\omega_m$ grows (see Appendix A). These methods, referred to as *RoPE scaling*, still require additional finetuning on long sequences, and don't generalize to long-context downstream tasks out of the box (Lu et al., 2024b).

**NoPE transformers.** In a parallel line of work, there have been efforts to train transformers without positional embeddings, commonly referred to as NoPE architectures Haviv et al. (2022); Kazemnejad et al. (2023) to avoid the need for rescaling RoPE frequencies. While NoPE was shown to be a viable LM architecture, it has failed to gain traction due to degraded performance (Haviv et al., 2022; Yang et al., 2025b). For an in-depth introduction to the above concepts, see Appendix A.

## 3 EXPLICIT POSITIONAL EMBEDDINGS ARE BENEFICIAL FOR TRAINING

While NoPE transformers were shown to be *expressive* enough for effective sequence modeling (Haviv et al., 2022; Kazemnejad et al., 2023), we find that they *consistently underperform state-of-the-art RoPE architectures* throughout our experiments. As illustrated in Figure 3, NoPE transformers maintain visibly worse perplexity *throughout training*. These empirical results are consistent with past literature (Haviv et al., 2022; Yang et al., 2025b), yet the reasons why positional embeddings are key for effective language model *training* have never been fully understood.

From a purely mechanistic perspective, even without explicit positional embeddings, NoPE transformers can exploit the causal mask to *encode* positional information, maintaining the same expressivity as their RoPE counterparts (Haviv et al., 2022; Kazemnejad et al., 2023). Specifically,

---

[1]Note the softmax in Equation 1 is taken on the first $i$ tokens, implementing a causal mask.

Kazemnejad et al. (2023) prove that the first attention layer in a NoPE transformer can *perfectly reconstruct* sequence positions, and subsequent layers can emulate the effects of relative or absolute positional embeddings. As detailed in Section 3.1, rather than looking at theoretical expressivity, we investigate this empirical performance discrepancy from a *training* perspective, providing theoretical analysis of the positional bias of NoPE transformers during optimization. The theoretical and empirical analysis in this section can be summarized in the following observation.

> **Observation 1.** *Positional information and attention non-uniformity, which are crucial for sequence modeling, develop at a **bounded rate** in NoPE transformers. In contrast, explicit PE methods, such as RoPE, provide a strong bias from the outset and facilitate fast propagation of positional information, resulting in faster training.*

At a high level, our analysis focuses on the rate at which NoPE and RoPE transformers can develop *positional bias* in their self-attention heads, which captures their non-uniformity. We quantify attention positional bias as a linear functional on the attention map:

> **Definition 3.1** (Attention positional bias). Given centered positional weights $c_{ij} \in \mathbb{R}$ with $\sum_{j \leq i} c_{ij} = 0$, the *positional bias* of the attention weights $\alpha_{ij}$ is
> $$\mathbf{A}^c(\alpha) = \frac{1}{T} \sum_{i=1}^{T} \sum_{j \leq i} c_{ij} \alpha_{ij}.$$

Attention heads with a strong positional bias would maximize the average value of $\mathbf{A}^c$ across input sequences. For example, a "diagonal" attention head, focusing mass on the current token, is exactly the maximizer of $\mathbf{A}^c$, with $c_{ij}$ having 1s on the diagonal and $-\frac{1}{i-1}$ otherwise.

To validate the theory behind Observation 1, we empirically compare the gradients of the attention positional bias functional in attention heads of RoPE and NoPE transformers. Specifically, we measure the average gradient norm at initialization in the direction of two common language modeling patterns: diagonal attention heads, placing mass on the current token, and off-diagonal heads, capturing immediate previous token context. As illustrated in Figure 4, the gradient magnitudes of NoPE transformers are far lower than those of RoPE transformers, with the gap between the two growing in deeper layers. This reflects NoPE's scalability challenges and difficulty in recovering positional information, which we theoretically analyze in the next section.

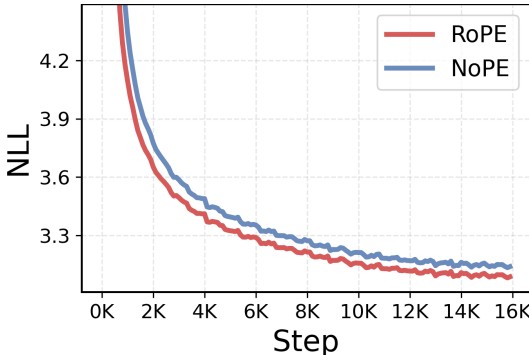

Figure 3: **RoPE outperforms NoPE.** Training loss curves for a RoPE and NoPE transformers on 16B fineweb tokens. RoPE outperforms NoPE *throughout training*.

## 3.1 Theoretical analysis

We detail our findings, summarized in Observation 1, with a series of formal results, bounding the rate at which positional bias can develop early in training. We provide full proofs and an extended analysis of these results in Appendix B. Throughout this section, we study the sensitivity of the attention positional bias $\mathbf{A}^c$ to the transformer's parameters and interpret $\|\nabla_\theta \mathbf{A}^c\|$ as bounding the rate at which non-uniform attention patterns can emerge during training.

**Warm-up: NoPE transformers break on constant sequences.** Before moving to the main theoretical result, we consider a motivating example that illustrates NoPE transformers' training difficulties. Because attention forms a convex combination of value vectors, an attention head applied to a sequence of identical tokens $x_1 = \cdots = x_T$ produces identical outputs at every position. Moreover, since normalization layers, MLP blocks, and residual connections act *pointwise* on tokens, this uniformity propagates through the network. In a NoPE transformer, this means

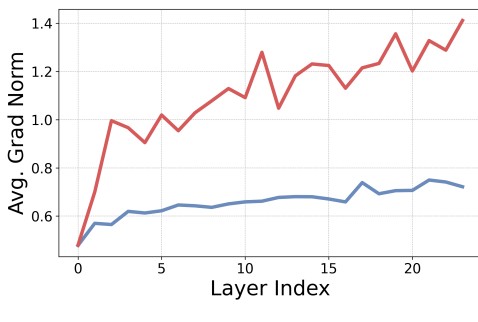
(a) Diagonal head bias.

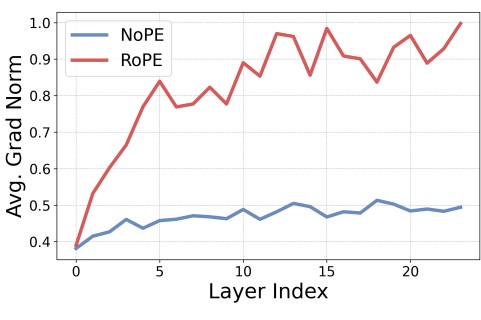
(b) Off-diagonal head bias.

Figure 4: **RoPE transformers have higher positional bias gradients at initialization.** We compare the average norm of $\mathbf{A}^c$ across layers, for RoPE and NoPE transformers. In 4a we plot the gradient norms of positional bias towards a diagonal head, and in 4b, we take bias towards previous token attention, off-diagonal head. In both cases, the gradient norm is consistently higher for RoPE across layers, meaning that RoPE heads can learn these patterns faster.

the attention logits are constant over all $j \leq i$, hence the post-softmax attention probabilities are uniform. Consequently, the model cannot induce any positional preference and $\mathbf{A}^c \equiv 0$ for *any* positional weights $c$.

> **Proposition 3.2.** *Let* M *be a NoPE transformer. If the input sequence* $x = (x_1, \ldots, x_T)$ *is comprised of identical tokens* $x_1 = \cdots = x_T$, *then (1)* ***all*** *attention heads are uniform:* $\alpha_{ij} = \frac{1}{i}$, *(2) query and key gradients vanish:* $\partial \mathcal{L}/\partial W_Q = \partial \mathcal{L}/\partial W_K = 0$, *(3) for all heads and any positional weights* $\mathbf{A}^c = 0$, $\nabla_\theta \mathbf{A}^c = \mathbf{0}$, *and (4) the output is constant:* $\mathsf{M}(x)_1 = \cdots = \mathsf{M}(x)_T$.

The explicit positional information injected into attention heads in RoPE transformers circumvents this issue. Enabling non-zero $\mathbf{A}^c$ gradients even on constant sequences.

> **Proposition 3.3.** *For a non-trivial RoPE attention head, even if the input sequence is constant, there are positional weights* $c$, *for which* $\mathbf{A}^c > 0$, *and* $\|\nabla_\theta \mathbf{A}^c\| > 0$.

**NoPE transformers propagate embedding uniformity.** At initialization, the entries of the embedding matrix are drawn i.i.d. from a distribution with a fixed small variance (commonly, $\sigma^2 = 0.02$). Therefore, the token embeddings are close to uniform at the beginning of training. The next theorem shows that for NoPE transformers, this uniformity persists throughout the network, and bounds the attention positional bias $\mathbf{A}^c$ and its gradients.

> **Theorem 3.4.** *Define the he prefix-spread of the hidden states at layer* $l$ *as*
> $$\Delta_h^{(l)} := \max_{1 \leq j \leq i \leq T} \left\| \bar{h}_i^{(l)} - h_j^{(l)} \right\|, \quad \text{where} \quad \bar{h}_i^{(l)} := \frac{1}{i} \sum_{j \leq i} h_j^{(l)}.$$
>
> *For NoPE transformers, there exists* $\varepsilon > 0$ *and constants* $C_1$, $C_2$, *and* $C_3$ *such that if the initial embeddings* $\Delta_h^{(1)} \leq \varepsilon$, *then for all layers* $l \leq L$:
>
> $$\Delta_h^{(l)} \leq C_1 \varepsilon, \qquad \left| \mathbf{A}^c \right| \leq C_2 \varepsilon, \qquad \left\| \partial \mathbf{A}^c / \partial W_Q \right\|, \left\| \partial \mathbf{A}^c / \partial W_K \right\| \leq C_3 \varepsilon,$$
>
> *with high probability over the initialization distribution. The constants only depend on the number of layers and heads, and* ***not*** *on the sequence length.*

The main idea in the proof of Theorem 3.4 is that uniformity in the embeddings causes uniformity in the attention maps, so $\alpha_{ij} \approx 1/i$. Uniform mixing of tokens cannot increase the prefix spread; thus, uniformity persists throughout the network. This result explains the discrepancy between RoPE and NoPE transformers illustrated in Figure 4.

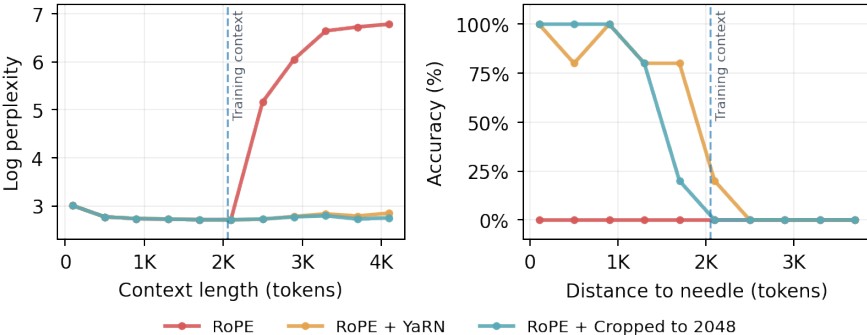

Figure 5: **YaRN crops effective retrieval context.** We compare RoPE's and YaRN's perplexity and NIAH performance at up-to $2\times$ the original context length against a baseline that *crops* the input sequence to the training context length. Both YaRN and the cropped baseline can maintain perplexity on sequences exceeding the training context length, but are unable to retrieve information placed far away from the query.

In summary, we demonstrate that while NoPE attention can learn positional bias, attention non-uniformity develops slowly early in training due to bounded $\mathbf{A}^c$ gradients at initialization.

## 4 RoPE PREVENTS EFFECTIVE ZERO-SHOT CONTEXT EXTENSION

State-of-the-art RoPE scaling methods fail to effectively generalize to sequences longer than those seen in training without additional long-context finetuning. While YaRN and other popular frequency scaling techniques do avoid perplexity degradation on long-context sequence (bloc97, 2023; Peng et al., 2023), they exhibit sharp performance drops on downstream tasks whenever important information is present deep in the sequence, beyond the training context (Lu et al., 2024b; Liu et al., 2023b). We empirically demonstrate this phenomenon, comparing the perplexity and needle-in-a-haystack (NIAH) (Kamradt, 2023; Hsieh et al., 2024) performance of a RoPE transformer scaled with YaRN and to a cropped context baseline. As illustrated in Figure 5, YaRN's zero-shot

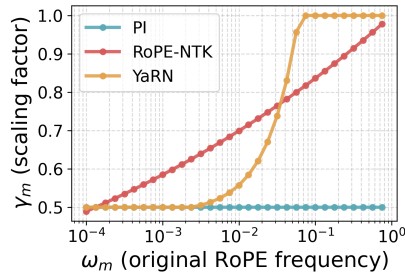

Figure 6: RoPE frequency scaling under PI, NTK-aware scaling (RoPE-NTK), and YaRN, with scaling factor $s = 2$.

behavior closely matches that of simply *cropping* the sequence length to the pretraining context, maintaining constant perplexity but ignoring information present outside the cropped window.

The cause of this limitation lies in the way context extension methods scale different RoPE frequencies. As detailed in Section 2, elaborated on in Appendix A, and illustrated in Figure 6, the scaling factors of PI (Chen et al., 2023), RoPE-NTK (bloc97, 2023), and YaRN (Peng et al., 2023) have a strong effect on *low frequencies*. In Section 4.1, we discuss why this scaling leads to the observed failures, yielding our second observation.

> **Observation 2.** *RoPE-scaling methods **must** compress low frequencies to keep positional phases in-distribution. This, in turn, shifts semantic attention heads at large relative distances, causing the observed failures on downstream tasks, preventing zero-shot context extension.*

### 4.1 WHY EXTRAPOLATION FAILURE IS INEVITABLE

**Effect of RoPE scaling.** RoPE scaling methods modify the frequencies at inference time to evaluate sequences that are longer than those seen during pretraining. In each $(2m, 2m+1)$ subspace, the RoPE phase at relative distance $\Delta$ is $\phi_m(\Delta) = \omega_m \Delta$, so scaling the frequency to $\omega'_m = \gamma_m \omega_m$ is equivalent to using a phase $\phi'_m(\Delta) = \gamma_m \omega_m \Delta$. As illustrated in Figure 6, most scaling methods

RoPE

| | 504 | 505 | 506 | 507 | 508 | 509 | 510 | 511 | 512 | 513 | 514 | 515 | 516 | 517 | 518 | 519 |

YaRN

| | 504 | 505 | 506 | 507 | 508 | 509 | 510 | 511 | 512 | 513 | 514 | 515 | 516 | 517 | 518 | 519 |

Key position

Figure 8: **RoPE scaling shifts semantic attention mass.** Attention weights of the last token (query) with tokens from a retrieval target (keys) in a semantic head evaluated on a NIAH probe. Since the head uses low frequencies and the relative distance is non-trivial, the impact of YaRN is substantial, shifting attention mass between tokens.

leave high frequencies nearly unchanged ($\gamma_m \approx 1$) but *all of them* compress the low frequencies ($\gamma_m \approx 1/s$). As demonstrated both theoretically and empirically in Barbero et al. (2024), high RoPE frequencies are primarily used by *positional heads*, with attention patterns based on relative token positions (e.g., diagonal or previous-token heads), whereas low frequencies are predominantly used by *semantic heads* that attend based on query/key content. Consequently, positional heads are largely unaffected by scaling, but semantic attention is shifted. Moreover, the effect on low-frequency dominated semantic heads is exacerbated for distant tokens, since the relative phase $\phi_m(\Delta)$ is larger, and thus the $1/s$ scaling factor has a greater effect. In other words, scaling *warps* low-frequency phases, shifting long-range attention in precisely the subspaces most used for semantic matching.

In Figure 7 and Figure 8, we illustrate this behavior in practice. We start by selecting a positional attention head in a pretrained QWEN2.5-0.5B model by examining its average attention positional bias (Definition 3.1) across layers. In Figure 7, we show the average attention weights in this positional head under YaRN scaling with $s = 2$. Because high frequencies, which are least affected by YaRN, dominate positional heads, the average attention profiles are similar. In Figure 8, we then contrast this behavior with that of a semantic head for a long

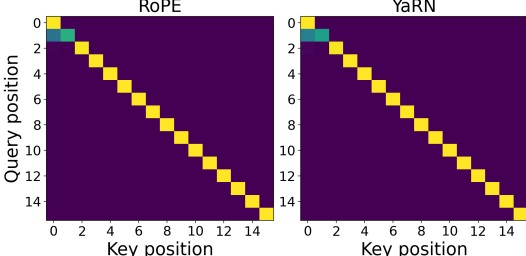

Figure 7: **RoPE scaling preserves average attention in positional heads.**

needle-in-a-haystack sequence, plotting the average attention of the last token (query) with tokens around the needle (keys). YaRN's aggressive scaling of low frequencies substantially shifts attention mass across tokens, reflecting the impact of frequency compression at longer ranges.

**Why this is inevitable.** In a standard RoPE setup, low-frequency phases never make a full cycle over the original context length: $\phi_m(C_{\text{train}}) = \omega_m C_{\text{train}} < 2\pi$ for small $\omega_m$. E.g. for a standard RoPE base $b = 10^4$, a transformer with head dimension $d_k = 64$, will have at least five low frequencies for which $\phi_m(C_{\text{train}}) < 2\pi$, even at a training context of $C_{\text{train}} = 32,000$. If we leave $\omega_m$ unchanged at an extended length $C_{\text{test}} > C_{\text{train}}$, the new maximal relative phase $\phi_m(C_{\text{test}})$ is pushed outside the training regime and becomes out of distribution for the head. Therefore, to constrain phases to remain in range, any scaling method must choose $\gamma_m \leq \frac{C_{\text{train}}}{C_{\text{test}}} = \frac{1}{s}$, which becomes increasingly small as the extension factor $s$ grows. In other words, when applying a RoPE transformer to sequences longer than those seen in training, any post-hoc scaling method *must* compress the low frequencies. But this compression, in turn, shifts attention weights at long relative distances.

## 5 DROPE: DROPPING POSITIONAL EMBEDDINGS AFTER PRETRAINING

Taken together, Observations 1 and 2 imply that providing explicit positional information with PE is a key component for effective LM training, but is also a fundamental barrier to long-context generalization. This raises a natural question: is it possible to harness the inductive bias from positional embeddings *exclusively* during pretraining? We answer in the affirmative. In this section, we empirically show that it is possible to drop all positional embeddings from a pretrained transformer and quickly recover the model's in-context capabilities with a short recalibration phase. Most notably, this simple new procedure (DroPE) unlocks strong *zero-shot* long context generalization to unseen sequence lengths, far beyond highly-tuned RoPE extensions and prior alternative architectures.

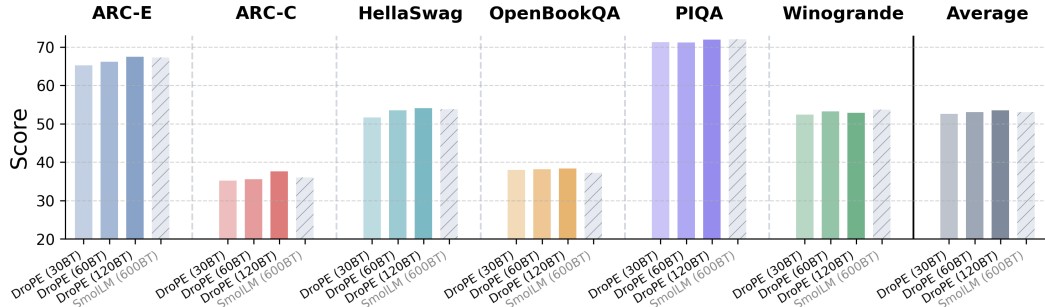

Figure 9: **DroPE matches base model in-context performance.** Comparison of base SMOLLM with SMOLLM-DROPE on standard LM benchmarks, using three recalibration recipes.

> **Observation 3.** *Positional embeddings can be **removed after pretraining**, allowing LMs to generalize **zero-shot** to **unseen sequence lengths** without compromising their in-context performance after short recalibration on a fraction of the training tokens at the original context size.*

## 5.1 LARGE-SCALE EMPIRICAL EVALUATION

We extensively validate DroPE with two different LMs and dataset scales, showing it outperforms prior approaches both as a *zero cost* integration into pretraining recipes and as an inexpensive way to adapt *any LM in the wild* already pretrained on hundreds of billions of tokens. For all experiments in this paper, we provide full implementation details of each evaluated architecture and optimization phase, including comprehensive hyperparameter lists in Appendix C.

**Integrating DroPE at no extra cost.** For our first set of experiments, we train from scratch different LMs with half a billion parameters on 16B fineweb tokens (Penedo et al., 2024), over twice the chinchilla-optimal rate (Hoffmann et al., 2022). We repeat this recipe for RoPE and NoPE transformers, as well as an RNoPE-SWA model Yang et al. (2025b), an alternative architecture specifically aimed at long-context capabilities. We implement DroPE by taking the 14B tokens RoPE transformer checkpoint, removing positional embeddings from every layer, and resuming training for the final 2B tokens. Despite only recalibrating at the very end of training, at no extra cost, DroPE matches the final in-context validation perplexity of RoPE trained on the full 16B tokens, showing a clear edge over the NoPE baseline trained without positional embedding all the way (Figure 2).

To evaluate the long-context generalization of each method, we select three tasks from the RULER benchmark (Hsieh et al., 2024): (1) *multi-query:* retrieve needles for several listed keys, (2) *multi-key:* retrieve the needle for one specified key, and (3) *multi-value:* retrieve all needles for one key with a single query. For the base RoPE transformer, we consider three context extension strategies: PI (Chen et al., 2023), NTK-RoPE (bloc97, 2023), and the popular YaRN (Peng et al., 2023) described in Section 2 and Appendix A.

Table 1: **Zero-shot NIAH at** $2\times$ **training context.** Results are reported as a success rate over 500 trials.

| Method | Multi-Query | Multi-Key | Multi-Value |
|---|---|---|---|
| RoPE transformer | 0.0 | 0.0 | 0.0 |
| RoPE transformer + PI | 0.0 | 0.0 | 0.0 |
| RoPE transformer + RoPE-NTK | 21.1 | 19.4 | 16.5 |
| RoPE transformer + YaRN | 17.8 | 0.5 | 14.6 |
| ALiBi transformer | 5.2 | 0.0 | 1.1 |
| NoPE transformer | 9.2 | 36.2 | 21.4 |
| RNoPE-SWA transformer | 5.2 | 25.6 | 20.6 |
| DroPE transformer | **28.0** | **41.6** | **23.3** |

For our DroPE and NoPE transformers, we follow Wang et al. (2024) and scale the softmax temperature at test time; we report the exact scaling scheme in Appendix C. 'In Table 1, we report the success rate on each task at $2\times$ the training context length. Our *DroPE transformer substantially outperforms all of our baselines* in each setting. While RoPE-NTK and YaRN also yield improvements to the original RoPE transformer, they consistently trail DroPE, as most evident on the multi-key task. In contrast, specialized architectures such as RNoPE-SWA (Yang et al., 2025b), ALiBi (Press et al., 2021), and NoPE Kazemnejad et al. (2023) underperform on multi-query tasks, which are the logic-intensive setting where strong base models excel. We believe these results provide compelling evidence toward validating DroPE's potential to be integrated as a standard component in the training pipeline of future generations of LMs.

Table 2: **DroPE outperforms RoPE-scaling methods on long context-tasks.** We evaluate SMOLLM-DROPE and the base SMOLLM model, extended with different RoPE scaling methods, on four long context language modeling tasks from Bai et al. (2023) and needle-in-a-haystack.

| Method | MultiFieldQA | MuSiQue | GovReport | LCC | NIAH | Avg. |
|---|---|---|---|---|---|---|
| SMOLLM | 4.03 | 0.4 | 4.48 | 5.99 | 0.0 | 2.98 |
| SMOLLM + PI | 13.68 | 2.45 | 5.67 | 11.52 | 0.0 | 6.66 |
| SMOLLM + RoPE-NTK | 18.87 | 4.89 | **23.71** | 8.26 | 29.84 | 17.11 |
| SMOLLM + YaRN | 20.78 | 4.77 | 15.03 | 10.87 | 48.25 | 19.94 |
| SMOLLM-DROPE | **29.33** | **7.93** | 21.87 | **18.56** | **74.92** | **30.52** |

**Extending the context of LMs in the wild with DroPE.** For our second set of experiments, we directly apply DroPE to a recent language model from the SMOLLM family (Allal et al., 2024) that was pretrained on over 600B tokens. We perform DroPE's recalibration for this model with continued pretraining using the same context length, data, and hyperparameters as reported by Allal et al. (2024). We consider three different recalibration budgets of 30, 60, and 120 billion tokens, adjusting the learning rate schedule accordingly. In order to support training at a high learning rate as perscribed by Allal et al. (2024), after applying DroPE we add QKNorm (Henry et al., 2020). We note that this addition not change the capacity of the our model, as analyzed in Appendix D.

We start by analyzing how quickly SMOLLM-DROPE can recover SMOLLM's in-context performance on six different LM reasoning benchmarks (Clark et al., 2018; Zellers et al., 2019; Mihaylov et al., 2018; Bisk et al., 2020; Sakaguchi et al., 2021). As shown in Figures 9 and 10 as well as Table 6, our shortest training schedule almost matches SMOLLM on every task while our longest schedule even manages to *exceed* its original performance. Furthermore, inspecting our model at every checkpoint throughout training, we find that DroPE quickly recovers *over 95% of* SMOLLM*'s performance* after less than 5B tokens, representing a minuscule $0.8\%$ of SMOLLMs original budget.

We then evaluate SMOLLM-DROPE's zero-shot length generalization on four different tasks from Long-Bench (Bai et al., 2023), a challenging benchmark even for closed-source LMs, including knowledge-extraction problems longer than *80 times* SMOLLM's pretraining context (2048 tokens). We compare our method with the base SMOLLM and three RoPE extensions: PI, RoPE-NTK, and YaRN. As shown in Table 2, despite a significant difficulty spike compared to our prior evaluations, DroPE still displays a clear edge over prior approaches, improving the base SMOLLM's average score by over 10 times. These gains are far beyond all prior zero-shot RoPE extensions currently used across modern LMs. Additionally, we evaluated SMOLLM-DROPE or needle-in-a-haystack tasks of size up to $8\times$ SMOLLM's original context length.

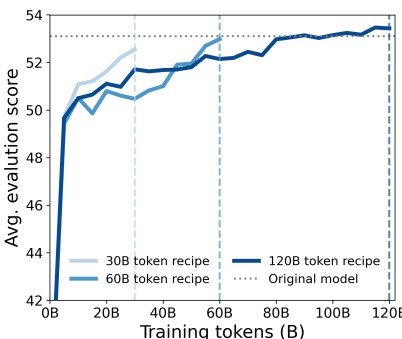

Figure 10: **SMOLLM-DROPE recalibration.** We compare three recipes, using 30B, 60B, and 120B training tokens.

Table 3: DroPE outperforms RoPE-scaling methods on long needle-in-a-haystack tasks.

| Method | $2\times$ original context | $4\times$ original context | $8\times$ original context |
|---|---|---|---|
| SMOLLM + RoPE-NTK | 29.84 | 14.37 | 7.19 |
| SMOLLM + YaRN | 48.25 | 25.62 | 12.18 |
| SMOLLM + LongRoPE2 | 44.20 | 26.20 | 16.45 |
| SMOLLM-DROPE | **74.92** | **55.00** | **52.20** |

Overall, our comprehensive in-context and out-of-context results provide a clear demonstration of a new, efficient, and effective long-context adaptation, which we believe can have concrete implications for reducing training costs and tackling the canonical context scalability challenges of transformers.

Table 4: **Length generalization results on larger models.** We evaluate DroPE on SMOLLM-1.7B and LLAMA2-7B, and compare it against different RoPE scaling methods, on long context language modeling tasks from Bai et al. (2023).

| Model | Method | MultiFieldQA | MuSiQue | GovReport | Avg. |
|---|---|---|---|---|---|
| SMOLLM-1.7B | Base | 4.12 | 0.50 | 4.70 | 3.11 |
| | RoPE-NTK | 27.58 | 3.37 | 24.65 | 18.53 |
| | YaRN | 27.60 | 3.90 | 17.19 | 16.23 |
| | DroPE | **32.18** | **7.53** | **24.77** | **21.49** |
| LLAMA2-7B | Base | 17.26 | 10.43 | 32.41 | 20.03 |
| | RoPE-NTK | 21.81 | 10.91 | 32.91 | 21.88 |
| | YaRN | 23.13 | 7.65 | 26.65 | 19.14 |
| | DroPE | **25.90** | **12.88** | **39.47** | **26.08** |

**Scaling to larger models.** To test DroPE's ability to scale to larger LMs in the wild we additionally apply DroPE to SMOLLM-1.7B (Allal et al., 2024) and LLAMA2-7B (Touvron et al., 2023). For both of these models, we perform recalibration on 20B tokens. For SMOLLM-1.7B, this represents 2% of the pretraining budget and for LLAMA2-7B the recalibration represents only 0.5% of the pretraining budget. As demonstrated in Table 4, even with this small relative recalibration budget, SMOLLM-1.7B-DROPE and LLAMA2-7B-DROPE outperform state-of-the-art RoPE-scaling methods on long-context question-answering and summarization. For additional experimental results, including results on the entire LongBench benchmark, and a performance by query length breakdown, see Appendix D.

## 6 RELATED WORK

Recent improvements to RoPE include variants based on Fourier and wavelet transforms (Hua et al., 2025; Oka et al., 2025) and methods such as $p$-RoPE (Barbero et al., 2025), NRoPE-SWA (Yang et al., 2025b), and SWAN-GPT (Puvvada et al., 2025) which occupy a middle ground between RoPE and NoPE. Our approach represents a fundamentally different paradigm, replacing RoPE with NoPE at different stages of training. These directions are complementary to ours and can be used in place of RoPE within the DroPE framework. A parallel direction seeks length generalization while retaining a dedicated positional vector yet modifying its indexing or adaptivity (zican Dong et al., 2024; Wu et al., 2024; Zheng et al., 2024). Another line of related works considers post-training architectural modification, e.g., for efficient inference (Ji et al., 2025).

## 7 DISCUSSION AND EXTENSIONS

Our findings support a reinterpretation of positional embeddings in transformer LMs as a useful inductive bias that is essential for efficient training (Observation 1), but inherently constrains zero-shot context extension (Observation 2). Based on these findings, we propose DroPE, a new method rethinking the conventional role of PEs as a temporary scaffold that can and should be removed after serving their training-time purpose (Observation 3). We empirically validate DroPE across different models and data scales, showing its effectiveness and potential to be integrated as a new core component of future state-of-the-art training pipelines. Integrating DroPE in state-of-the-art training pipelines for autoregressive or diffusion-based LMs is an interesting direction for future work. More broadly, our work demonstrates that canonical trade-offs in LM design can be reconciled by employing different architectural choices for training and inference, which we hope will inspire further research toward challenging established bottlenecks in AI.

## ETHICS STATEMENT

DroPE aims to further advance the field of long context language modeling by removing positional embeddings after pretraining. Our intended use is to advance robust, efficient, long-context reasoning in open research environments and practical systems. Our experiments use publicly available training corpora and benchmarks. Given this, we foresee no issues regarding fairness, privacy, or security, or any other harmful societal or ethical implications in general.

## Reproducibility statement

We provide our full source code in the supplementary material and provide all details of our experimental setup – including datasets, model specification, training regime, evaluation protocol, and the full set of hyperparameters – in Appendix C. Our base models and datasets are all publicly available.

## Acknowledgments

YG is supported by the UKRI Engineering and Physical Sciences Research Council (EPSRC) CDT in Autonomous and Intelligent Machines and Systems (grant reference EP/S024050/1).

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

## A EXTENDED PRELIMINARIES

**Attention.** Throughout this section, we consider a pre-norm, decoder-only transformer with $L$ layers, $H$ attention heads per layer, model dimension $d = d_{\text{model}}$, and head dimension $d_k$. $h_1^{(l)}, \ldots, h_T^{(l)} \in \mathbb{R}^d$ denote the representations fed into the $l$-th multi-head attention block. For a head $h$ in layer $l$, queries, keys, and values are computed by

$$q_i^{(l,h)} = W_Q^{(l,h)} h_i^{(l)}, \qquad k_i^{(l,h)} = W_K^{(l,h)} h_i^{(l)}, \qquad v_i^{(l,h)} = W_V^{(l,h)} h_i^{(l)}, \tag{4}$$

The attention scores and weights are then computed by

$$s_{ij}^{(l,h)} = \frac{1}{\sqrt{d_k}} \big(q_i^{(l,h)}\big)^\top k_j^{(l,h)}, \qquad \alpha_{ij}^{(l,h)} = \text{softmax}(s_{i1}^{(l,h)}, \ldots, s_{ii}^{(l,h)})_j. \tag{5}$$

$s_{ij}^{(l,h)}$ are referred to as attention *logits* or *scores* and $\alpha_{ij}^{(l,h)}$ are referred to as attention *weights* or *probabilities*. Note that the softmax is taken over $j \leq i$, implementing a causal mask. The output of the multi-head attention block is

$$z_i^{(l,h)} = \sum_{j \leq i} \alpha_{ij}^{(l,h)} v_j^{(l,h)}, \qquad o_i^{(l)} = W_O^{(l)} [z_i^{(l,1)}, \ldots, z_i^{(l,H)}], \tag{6}$$

where $[\cdot, \ldots, \cdot]$ represents concatenation along the feature dimension. When clear from context, we omit layer and head indices.

**Positional embeddings in transformers.** The attention mechanism does not directly encode relative distances between queries and keys. Therefore, attention is invariant to prefix permutations: for any permutation $\sigma \in S_p$ of the first $p$ input tokens, $\text{attn}(x_{\sigma^{-1}(1)}, \dots, x_{\sigma^{-1}(p)}, x_p, \dots, x_T)_i = \text{attn}(x_1, \dots, x_T)_i$ for every $i > p$. In other words, pure attention is *blind* to token positions. To address this, Vaswani et al. (2017) introduced *absolute* positional embeddings, adding position information to the token embeddings before the first transformer block. More recently, many architectures replace absolute embeddings with *relative* schemes that inject pairwise positional information directly into the attention mechanism. The most widely used approach is Rotary Position Embedding (RoPE) (Su et al., 2024). RoPE modifies the attention scores in Equation 5 by rotating queries and keys before taking their inner product:

$$s_{ij}^{\text{RoPE}} = \frac{1}{\sqrt{d_k}} q_i^\top R^{j-i} k_j, \qquad \alpha_{ij}^{\text{RoPE}} = \text{softmax}(s_{i1}^{\text{RoPE}}, \cdots, s_{ii}^{\text{RoPE}})_j, \tag{7}$$

where, $R \in O(d_k)$ is a block-diagonal orthogonal matrix composed out of $2 \times 2$ rotation blocks:

$$R = \text{block-diag}\left(R(\omega_1), \dots, R(\omega_{d_k/2})\right), \quad R(\omega) = \begin{pmatrix} \cos(\omega) & -\sin(\omega) \\ \sin(\omega) & \cos(\omega) \end{pmatrix}. \tag{8}$$

In the standard RoPE parameterization, $\omega_m = b^{-2\frac{m-1}{d_k}}$ with $b = 10{,}000$.

**Language model context extension.** Generalizing to contexts longer than those seen during training is a key challenge for transformer-based language models. The key issue is that when applying a transformer on a longer context, the attention mechanism must operate over more tokens than it was trained to handle. This issue is exacerbated with RoPE: applying RoPE to sequences beyond the training length introduces larger position deltas, and thus larger rotations, pushing attention logits out of the training distribution. RoPE context-extension methods address this by *rescaling the RoPE frequencies* when the inference context length exceeds the training context length. Let $C_{\text{train}}$ be the training context and $C_{\text{test}} > C_{\text{train}}$ the target context with extension factor $s = C_{\text{test}}/C_{\text{train}}$. Such methods define new frequencies

$$\omega_m' = \gamma_m \omega_m, \qquad m = 1, \dots, \tfrac{d_k}{2},$$

using *scaling factors* $\gamma_m = \gamma_m(s)$. E.g. Position Interpolation (PI) (Chen et al., 2023), uses a uniform scaling of

$$\gamma_m^{\text{PI}} = \tfrac{1}{s}. \tag{9}$$

NTK-RoPE (bloc97, 2023) uses

$$\gamma_m^{\text{NTK}} = \left(\tfrac{1}{s}\right)^{\frac{2m}{d_k-2}}, \tag{10}$$

so that low frequencies ($m \approx d_k/2$) are scaled similarly to PI and for high frequencies $\gamma_m \approx 1$. YaRN (Peng et al., 2023) uses

$$\gamma_m^{\text{YaRN}} = (1 - \kappa_m)\tfrac{1}{s} + \kappa_m, \qquad \kappa_m = \begin{cases} 0 & \omega_m < p \\ 1 & \omega_m > q \\ \frac{\omega_m - p}{q - p} & p \le \omega_m \le q, \end{cases} \tag{11}$$

with tunable $p$ and $q$ parameters, originally chosen as $p = 1, q = 32$. See Figure 11 for a comparison between these different RoPE scaling methods with $s = 2, 3$, and 4.

## B  THEORETICAL RESULTS AND PROOFS

In this section, we analyze the behavior of positional bias, or attention non-uniformity, in NoPE transformers and RoPE transformers early in training. We provide formal statements and proofs for all the results from Section 3, starting with Propositions 3.2 and 3.3, followed by Theorem 3.4. The notation of this section follows that of Appendix A.

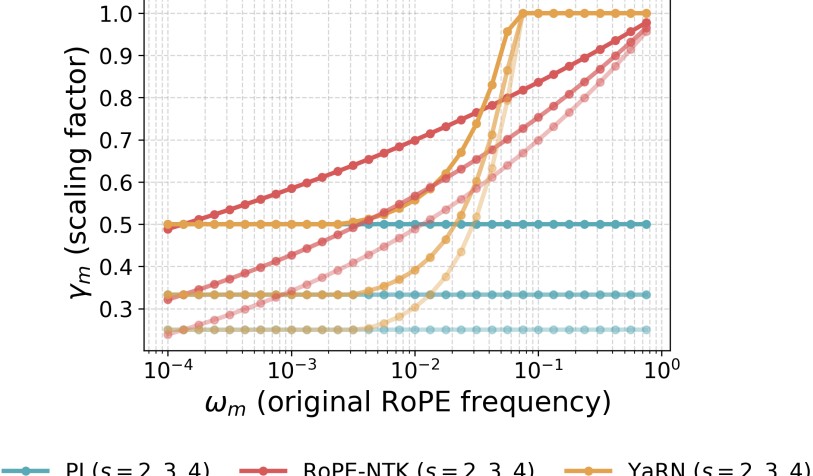

Figure 11: RoPE frequency scaling under PI, NTK-aware scaling (RoPE-NTK), and YaRN, with scaling factors $s = 2, 3, 4$.

## B.1 PROOF OF PROPOSITION 3.2

**Proposition 3.2.** *Let* M *be a NoPE transformer. If the input sequence* $x = (x_1, \ldots, x_T)$ *is comprised of identical tokens* $x_1 = \cdots = x_T$, *then (1)* **all** *attention heads are uniform:* $\alpha_{ij} = \frac{1}{i}$, *(2) query and key gradients vanish:* $\partial\mathcal{L}/\partial W_Q = \partial\mathcal{L}/\partial W_K = 0$, *(3) for all heads and any positional weights* $\mathbf{A}^c = 0$, $\nabla_\theta \mathbf{A}^c = \mathbf{0}$, *and (4) the output is constant:* $M(x)_1 = \cdots = M(x)_T$.

*Proof.* Let $x_1, \ldots, x_T$ be a constant input sequence, $x_1 = \cdots = x_T$, and let M be a NoPE transformer, i.e. a transformer with *no positional encodings* and causal self attention. The order of the proof is $(4) \Rightarrow (1) \Rightarrow (2 + 3)$.

**(4) Layer outputs, and thus model outputs, are constant.** At the first layer, inputs are identical $h_1^{(1)} = \cdots = h_L^{(1)} = h$. This means that for every attention head and every $1 \le j \le T$

$$v_j \equiv v = W_V h.$$

Therefore, the output of the attention head is

$$z_i = \sum_{j \le i} \alpha_{ij} v_j = \sum_{j \le i} \alpha_{ij} v = \left(\sum_{j \le i} \alpha_{ij}\right) v = 1 \cdot v,$$

independent of $i$. Concatenating heads and applying $W_O$ preserves equality across positions. Residual connections, LayerNorm, and the MLP are positionwise (the same function is applied independently at each position), so identical inputs produce identical outputs at every position. Thus the layer output remains constant. By repeating this argument layer-by-layer, every subsequent layer receives identical inputs and outputs identical states, so in the end

$$M(x)_1 = \cdots = M(x)_L.$$

**(1) Uniform causal attention.** Using (4), we know that for every layer $1 \le l \le L$

$$h_1^{(l)} = \cdots = h_L^{(l)} = h.$$

Therefore, for every attention head and every $1 \le j \le T$

$$q_j \equiv q := W_Q h, \quad k_j \equiv k := W_K h, \quad v_j \equiv v := W_V h.$$

Thus, for each $1 \le j \le i \le T$, the attention scores $s_{ij} = q^\top k / \sqrt{d_k} \equiv c$ are constant (independent of $i$ or $j$). Hence

$$\alpha_{ij} = \mathrm{softmax}(\underbrace{c, \ldots, c}_{i \text{ entries}})_j = \tfrac{1}{i} \qquad (j \le i).$$

**(2 + 3) Vanishing** $W_Q, W_K$ **gradients.** Since, the inputs for every layer are constant, we know from (1) that every attention head has $\alpha_{ij} \equiv 1/i$, *independant of* $W_Q$ *and* $W_K$. Therefore $\partial\alpha_{ij}/\partial W_Q = \partial\alpha_{ij}/\partial W_K = 0$. Since the attention bias $\mathbf{A}^c$ depends on the parameters $\theta$ only through $\alpha_{ij}$ and the loss $\mathcal{L}$ depends on $W_Q$ and $W_K$ only through $\alpha_{ij}$, all these gradients vanish. More formally, using the chain rule,

$$\frac{\partial\mathbf{A}^c}{\partial\theta} = \frac{1}{T}\sum_{1\leq j\leq i\leq T} c_{ij}\frac{\partial\alpha_{ij}}{\partial\theta} = 0,$$

$$\frac{\partial\mathcal{L}}{\partial W_Q} = \sum_{1\leq j\leq i\leq T}\frac{\partial\mathcal{L}}{\partial\alpha_{ij}}\frac{\partial\alpha_{ij}}{\partial W_Q} = 0, \qquad \frac{\partial\mathcal{L}}{\partial W_K} = \sum_{1\leq j\leq i\leq T}\frac{\partial\mathcal{L}}{\partial\alpha_{ij}}\frac{\partial\alpha_{ij}}{\partial W_K} = 0.$$

Additionally, since the heads are uniform the attention bias is zero to begin with

$$\mathbf{A}^c = \frac{1}{T}\sum_{1\leq j\leq i\leq T} c_{ij}\alpha_{ij} = \frac{1}{T}\sum_{i=1}^T \frac{1}{i}\sum_{j\leq i} c_{ij} = \frac{1}{T}\sum_{i=1}^T \frac{1}{i}\cdot 0 = 0.$$

$\square$

*Remark* B.1. Note that part (4) of the proposition holds for RoPE transformers as well. Parts (1), (2) and (3) *do not*. The relative rotations break attention uniformity and thus changing the magnitude of $\|W_Q\|$ and $\|W_K\|$ can affect the attention weights. This is formally demonstrated in the next section.

## B.2 Proof of Proposition 3.3

**Proposition 3.3.** *For a non-trivial RoPE attention head, even if the input sequence is constant, there are positional weights c, for which* $\mathbf{A}^c > 0$, *and* $\|\nabla_\theta\mathbf{A}^c\| > 0$.

*Proof.* Let $x_1 = \cdots = x_T = x \in \mathbb{R}^d$ be the inputs to a RoPE attention head, and let $W_Q, W_K \in \mathbb{R}^{d_k\times d}$ be the query and key projection parameters. Since the projection maps are shared across tokens, the queries and keys are constant as well:

$$q_i = W_Q x_i = W_Q x = q, \qquad k_i = W_K x_i = W_K x = k.$$

Set the positional bias weights to be

$$c_{ij} = \alpha_{ij} - \tfrac{1}{i}.$$

Since $\sum_{j\leq i}\alpha_{ij} = 1$, we have $\sum_{j\leq i} c_{ij} = 0$ as required. The positional bias $\mathbf{A}^c$ is

$$\mathbf{A}^c = \frac{1}{T}\sum_{i=1}^T\sum_{j\leq i}(\alpha_{ij}^2 - \tfrac{1}{i}\alpha_{ij}) = \frac{1}{T}\sum_{i=1}^T\Big(\sum_{j\leq i}\alpha_{ij}^2 - \tfrac{1}{i}\Big).$$

By Cauchy-Schwarz,

$$1 = \Big(\sum_{j\leq i}\alpha_{ij}\cdot 1\Big)^2 \leq \Big(\sum_{j\leq i}\alpha_{ij}^2\Big)\Big(\sum_{j\leq i}1\Big) = i\sum_{j\leq i}\alpha_{ij}^2,$$

with equality only when $\alpha_{i1} = \cdots = \alpha_{ii}$. Therefore,

$$\sum_{j\leq i}\alpha_{ij}^2 \geq \frac{1}{i},$$

with equality iff $\alpha_{ij} = 1/i$ is uniform. Therefore, $\mathbf{A}^c > 0$ unless $\alpha_{ij}$ is uniform for all $i$. The following lemma asserts that this is not the case

**Lemma B.2.** *For any non-degenerate RoPE head and input embeddings* $x_1 = \cdots = x_t = x$, *there exists* $i \geq 1$ *such that* $s_{i1}, \ldots, s_{ii}$ *and* $\alpha_{i1}, \ldots, \alpha_{ii}$ *are not uniform.*

The proof of Lemma B.2 is at the end of this subsection. As for $\nabla_\theta \mathbf{A}^c$, rewrite $\mathbf{A}^c$ as

$$\mathbf{A}^c = \frac{1}{T} \sum_{i=1}^{T} \left( \sum_{j \leq i} \alpha_{ij}^2 - \frac{1}{i} \right) = \frac{1}{T} \sum_{i=1}^{T} F_i - \frac{1}{T} \sum_{i=1}^{T} \frac{1}{i},$$

so the dependence in the parameters $\theta$ is entirely through

$$F_i := \sum_{j \leq i} \alpha_{ij}^2.$$

From the definition of RoPE, we have

$$\alpha_{ij} = \text{softmax}(s_{i1}, \ldots, s_{ii})_j, \qquad s_{ij} = \frac{1}{\sqrt{d_k}} q^\top R^{j-i} k.$$

Consider scaling $q$ by a scalar $\lambda > 0$: $q \mapsto \lambda q$. For fixed prefix $i$, define

$$Z_i(\lambda) := \sum_{j \leq i} e^{\lambda s_{ij}}, \qquad \alpha_{ij}(\lambda) = \frac{e^{\lambda s_{ij}}}{Z_i(\lambda)}, \qquad F_i(\lambda) := \sum_{j \leq i} \alpha_{ij}(\lambda)^2.$$

Then

$$F_i(\lambda) = \frac{Z_i(2\lambda)}{Z_i(\lambda)^2} \quad \Longrightarrow \quad \frac{d}{d\lambda} \log F_i(\lambda) = 2 \underbrace{\mathbb{E}_{j \sim \alpha_i(2\lambda)}[s_{ij}]}_{A_i'(2\lambda)} - 2 \underbrace{\mathbb{E}_{j \sim \alpha_i(\lambda)}[s_{ij}]}_{A_i'(\lambda)},$$

where $A_i(\lambda) := \log Z_i(\lambda)$ is the log-partition function. The second derivative of the log partition function is the logit variance

$$A_i''(\lambda) = \frac{Z_i''(\lambda) Z_i(\lambda) - (Z_i'(\lambda))^2}{(Z_i(\lambda))^2} = \frac{\sum_{j \leq i} s_{ij}^2 e^{\lambda s_{ij}}}{Z_i(\lambda)} - \left( \frac{\sum_{i \leq j} s_{ij} e^{\lambda s_{ij}}}{Z_i(\lambda)} \right)^2 = \text{Var}_{j \sim \alpha_i(\lambda)}(s_{ij})$$

therefore $A_i''(\lambda) = \text{Var}_{\alpha_i(\lambda)}(s_{i\cdot}) > 0$ since from Lemma B.2 $s_{ij}$ are not all equal and $\alpha_{ij}(\lambda) > 0$. Thus, $A_i'(\lambda)$ is strictly increasing in $\lambda$. Hence, for any $i$ with *non-constant* logits,

$$\frac{d}{d\lambda} F_i(\lambda) = F_i(\lambda) \cdot 2\big(A_i'(2\lambda) - A_i'(\lambda)\big) > 0,$$

and in particular at $\lambda = 1$,

$$\frac{d}{d\lambda} F_i(\lambda) \Big|_{\lambda=1} > 0.$$

By the chain rule for $q \mapsto \lambda q$,

$$\frac{d}{d\lambda} F_i(\lambda) \Big|_{\lambda=1} = \nabla_q F_i(q)^\top \cdot q.$$

Thus $\nabla_q F_i(q) \neq 0$ (otherwise the dot product with $q$ couldn't be strictly positive). Finally, since $q = W_Q x$,

$$\nabla_{W_Q} F_i = \nabla_q F_i x^\top,$$

and with $x \neq 0$ we get $\|\nabla_\theta F_i\| \geq \|\nabla_{W_Q}\| F_i > 0$. Therefore

$$\nabla_\theta \mathbf{A}^c = \frac{1}{T} \sum_{i=1}^{T} \nabla_\theta F_i$$

has strictly positive norm (a sum of nonzero matrices sharing the same nonzero right factor $x^\top$ cannot be the zero matrix unless all left factors vanish, which they don't for $i \geq 2$). $\qquad \square$

To conclude this section, we now prove Lemma B.2.

**Lemma B.2.** *For any non-degenerate RoPE head and input embeddings $x_1 = \cdots = x_t = x$, there exists $i \geq 1$ such that $s_{i1}, \ldots, s_{ii}$ and $\alpha_{i1}, \ldots, \alpha_{ii}$ are not uniform.*

*Proof.* RoPE acts as independent $2 \times 2$ rotations on disjoint coordinate pairs. Thus

$$R^\Delta = \bigoplus_{m=1}^M R(\Delta\omega_m), \quad M = d_k/2$$

with pairwise distinct frequencies $\omega_m \in (0, 2\pi)$. Decompose

$$q = (q_1, \ldots, q_M), \quad k = (k_1, \ldots, k_M), \qquad a_m, b_m \in \mathbb{R}^2,$$

so $s_{ij} = f(j - i)$ where

$$f(\Delta) = \frac{1}{\sqrt{d_k}} \sum_{m=1}^M q_m^\top R(\Delta \cdot \omega_m) k_m.$$

Let

$$R(\phi) = \begin{pmatrix} \cos\phi & -\sin\phi \\ \sin\phi & \cos\phi \end{pmatrix}, \qquad J = \begin{pmatrix} 0 & -1 \\ 1 & 0 \end{pmatrix}.$$

For any $u, v \in \mathbb{R}^2$,

$$u^\top R(\phi)\, v = (u^\top v) \cos\phi + (u^\top \cdot Jv) \sin\phi.$$

Define $A_m := q_m^\top k_m$ and $B_m := q_m^\top J b_m$. Then

$$f(\Delta) = \frac{1}{\sqrt{d_k}} \sum_{m=1}^M \Big( A_m \cos(\Delta\omega_m) + B_m \sin(\Delta\omega_m) \Big) = \Re\left( \sum_{m=1}^M c_m e^{i\Delta\omega_m} \right)$$

$$= \frac{1}{2} \sum_{m=1}^M C_m e^{i\Delta\omega_m} + \bar{C}_m e^{-i\Delta\omega_m},$$

where

$$C_m := \frac{A_m - iB_m}{\sqrt{d_k}}.$$

Assume $f(\Delta)$ is constant in $\Delta$ for $\Delta = 0, \ldots, 2M = d_k$, and denote the constant value by $-\frac{1}{2}C_0$. Then we have

$$\sum_{m=-M}^M C_m e^{i\Delta\omega_m} \equiv 0$$

were $C_{-m} := \bar{C}_m$, and $\omega_{-m} = -\omega_m$. Since $\{e^{-i\omega_M}, \ldots, e^{-i\omega_1}, 1, e^{i\omega_1}, \ldots, e^{i\omega_M}\}$ are all distinct, by Vandermonde's identity this means $C_m = \bar{C}_m = 0$ for $m = 1, \ldots, M, \Rightarrow A_m = B_m = 0$ for $m = 1, \ldots, M$. Now $A_m = B_m = 0$ means

$$q_m \perp k_m \quad \text{and} \quad q_m \perp Jk_m.$$

If $k_m \neq 0$, then $\{k_m, Jk_m\}$ spans $\mathbb{R}^2$, forcing $q_m = 0$. Thus for every block $m$, either $q_m = 0$ or $k_m = 0$, which results in a degenerate RoPE head, contradicting the assumption. Therefore, for $i \geq d_k + 1$ the attention logits $s_{ij}$ are not constant, and thus the attention weight $\alpha_{ij}$ are not constant. □

## B.3 PROOF OF THEOREM 3.4

In this section, we prove Theorem 3.4. To do so, we first need to prove a sequence of Propositions and Lemmas. First, we restate the theorem here.

**Theorem 3.4.** *Define the he prefix-spread of the hidden states at layer $l$ as*

$$\Delta_h^{(l)} := \max_{1 \leq j \leq i \leq T} \big\| \bar{h}_i^{(l)} - h_j^{(l)} \big\|, \quad where \quad \bar{h}_i^{(l)} := \frac{1}{i} \sum_{j \leq i} h_j^{(l)}.$$

*For NoPE transformers, there exists $\varepsilon > 0$ and constants $C_1$, $C_2$, and $C_3$ such that if the initial embeddings $\Delta_h^{(1)} \leq \varepsilon$, then for all layers $l \leq L$:*

$$\Delta_h^{(l)} \leq C_1\varepsilon, \qquad \big|\mathbf{A}^c\big| \leq C_2\varepsilon, \qquad \big\|\partial\mathbf{A}^c/\partial W_Q\big\|, \big\|\partial\mathbf{A}^c/\partial W_K\big\| \leq C_3\varepsilon,$$

*with high probability over the initialization distribution. The constants only depend on the number of layers and heads, and **not** on the sequence length.*

Since all weight matrices are drawn from a Gaussian distribution with a fixed variance, there exists a constant $B$, depending only on the architecture, such that with high probability the operator norms of $W_Q, W_K, W_V,$ and $W_O$, as well as the Lipschitz constants of the MLPs and normalization layers are all bounded by $B$. To see this use, e.g. Theorem 4.4.5 from Vershynin (2018) and the fact that for a two layer MLP $f$, it's Lipschitz constnat is bounded by $\text{Lip}(f) \leq \|W_1\| \|W_2\| \text{Lip}(\sigma)$. Let $L$ be the number of layers, and $H$ be the number of attention heads per layer. For any vector sequence $a_i \in \mathbb{R}^d$ we denote by $\bar{a}_i = \frac{1}{i} \sum_{j \leq i} a_j$ the prefix sum of $a_i$. For real sequences with two indices $a_{ij} \in \mathbb{R}$ we denote $a_i = (a_{i1}, \ldots, a_{ii}) \in \mathbb{R}^i$ and $\bar{a}_i = \frac{1}{i} \sum_{j \leq i} a_{ij}$.

**Proposition B.3.** *Fix a row $i$ in an attention head at the $l$-th layer.*

$$\max_{j \leq i} \left| s_{ij} - \bar{s}_i \right| \leq B^2 \sqrt{H} \Delta_h^{(l)}.$$

*Proof.* Notice that

$$s_{ij} - \bar{s}_i = \frac{1}{\sqrt{d_k}} q_i^\top k_j + \frac{1}{i} \sum_{r \leq i} \frac{1}{\sqrt{d_k}} q_i^\top k_r = \frac{1}{\sqrt{d_k}} q_i^\top (k_j - \bar{k}_i).$$

Therefore, by Cauchy-Swartz

$$\left| s_{ij} - \bar{s}_i \right| \leq \frac{1}{\sqrt{d_k}} \|q_i\| \|k_j - \bar{k}_i\|.$$

By the linearity of $W_K$ we get $\|k_j - \bar{k}_i\| = \|W_K(h_j - \bar{h}_i)\| \leq \|W_K\| \|h_j - \bar{h}_i\| \leq \|W_K\| \Delta_h^{(l)} \leq B\Delta_h^{(l)}$. As for $\|q_i\| = \|W_Q h_i\|$, recall that $h_i$ are the output of a normalization layer, and therefore (at initialization) $\|h_i\| = \sqrt{d}$. Thus, $\|q_i\| \leq B\sqrt{d}$. Putting it all together gives

$$\left| s_{ij} - \bar{s}_i \right| \leq B^2 \sqrt{\frac{d}{d_k}} \Delta_h^{(l)} = B^2 \sqrt{H} \Delta_h^{(l)}.$$

To finish the proof, take a maximum over $j \leq i$. $\qquad \square$

To bound the effect on the attention *probabilities*, we need the following Lemma.

**Lemma B.4.** *For any $b \in \mathbb{R}^n$,*

$$\|\text{softmax}(a + b) - \text{softmax}(a)\|_1 \leq \|b\|_\infty.$$

*Proof.* A $C^2$ convex function $f : \mathbb{R}^n \to \mathbb{R}$ satisfies $\|\nabla f(x) - \nabla f(y)\|_1 \leq \|x - y\|_\infty$ (1-smoothness) if $d^\top \nabla^2 f(x) d \leq \|d\|_\infty^2$ for all $x, d \in \mathbb{R}^n$ (see Theorem 2.1.6 in Nesterov (2013)). Take $f(x) = \log \left( \sum_{i=1}^n e^{x_i} \right)$. $f$ is $C^2$, convex and $\nabla f(x) = \text{softmax}(x)$. Therefore, all we need to show is that for all $x, d \in \mathbb{R}^n$

$$d^\top \nabla \text{softmax}(x) d = d \nabla^2 f(x) d \leq \|d\|^2.$$

and indeed,

$$\begin{aligned} d^\top \nabla \text{softmax}(x) d &= d^\top \text{diag}(\text{softmax}(x)) d - (\text{softmax}(x)^\top d)^2 \\ &\leq d^\top \text{diag}(\text{softmax}(x)) d \\ &\leq \|d\|_\infty^2 \|\text{softmax}(x)\|_1 \\ &= \|d\|_\infty^2, \end{aligned}$$

as required. $\qquad \square$

Using Lemma B.4, we can bound the uniformity of $\alpha_{ij}$ and the prefix spread of the head outputs.

**Proposition B.5.** *Let $u_i = \frac{1}{i} \mathbf{1} \in \mathbb{R}^i$. In any layer $l$,*

$$\|\alpha_i - u_i\|_1 \leq B^2 \sqrt{H} \Delta_h^{(l)}, \tag{12}$$

*and,*

$$\|z_i - \bar{v}_i\| \leq B^3 \sqrt{H} \left( \Delta_h^{(l)} \right)^2. \tag{13}$$

*Proof.* To get Equation 12, let $a$ be the constant vector $(\bar{s}_i, \ldots, \bar{s}_i) \in \mathbb{R}^i$ and let $b = s_i - a$. By Lemma B.4

$$\|\alpha_i - u_i\|_1 = \|\text{softmax}(a + b) - \text{softmax}(a)\|_1 \leq \|b\|_\infty.$$

Now, notice that $\|b\|_\infty = \max_{j \leq i} |s_{ij} - \bar{s}_i|$, therefore Proposition B.3 gives us the desired inequality. For Equation 13 notice that,

$$z_i - \bar{v}_i = \sum_{j \leq i} (\alpha_{ij} - \tfrac{1}{i})(v_j - \bar{v}_i),$$

hence

$$\|z_i - \bar{v}_i\| \leq \max_{j \leq i} \|v_j - \bar{v}_i\| \, \|\alpha_i - u_i\|_1 \leq B\Delta_h^{(l)} \cdot B^2 \sqrt{H} \Delta_h^{(l)} = B^3 \sqrt{H} \big(\Delta_h^{(l)}\big)^2.$$

$\square$

We now bound the next layer's spread in terms of the current one. Denote by $\Delta_z^{(l)} := \max_i \max_{j \leq i} \|z_j - \bar{z}_i\|$ the prefix spread of an attention head's output. First, we'll give a bound for $\Delta_z^{(l)}$, and then use this bound to prove the entire propagation result. Before, we need a short lemma.

**Lemma B.6.** *For any sequence $(x_j)$ and $j \leq i$,*

$$\|\bar{x}_j - \bar{x}_i\| \leq \max_{r \leq j} \|x_r - \bar{x}_i\| \leq \max_{r \leq i} \|x_r - \bar{x}_i\|.$$

*Proof.* $\bar{x}_j - \bar{x}_i = \frac{1}{j} \sum_{r \leq j} (x_r - \bar{x}_i)$ and triangle inequality. $\square$

**Proposition B.7.** *For any layer $1 \leq l \leq L$,*

$$\Delta_z^{(l)} \leq 2B\Delta_h^{(l)} + 2B^3 \sqrt{H} \big(\Delta_h^{(l)}\big)^2.$$

*Proof.* Fix $i$ and $j \leq i$. Write $z_j - \bar{z}_i = (\bar{v}_j - \bar{v}_i) + (z_j - \bar{v}_j) - (\bar{z}_i - \bar{v}_i)$, so

$$\|z_j - \bar{z}_i\| \leq \underbrace{\|\bar{v}_j - \bar{v}_i\|}_{=(a)} + \underbrace{\|z_j - \bar{v}_j\|}_{=(b)} + \underbrace{\|\bar{z}_i - \bar{v}_i\|}_{=(c)}.$$

By Lemma B.6,

$$(a) = \|\bar{v}_j - \bar{v}_i\| \leq \max_{r \leq i} \|v_r - \bar{v}_i\| \leq \|W_V\| \Delta_h^{(l)} \leq B\Delta_h^{(l)}.$$

By Proposition B.5

$$(b) \leq B^3 \sqrt{H} \big(\Delta_h^{(l)}\big)^2.$$

As for (c), Notice that,

$$\bar{z}_i - \bar{v}_i = \frac{1}{i} \sum_{r \leq i} (z_r - \bar{v}_i) = \frac{1}{i} \sum_{r \leq i} \big((z_r - \bar{v}_r) + (\bar{v}_r - \bar{v}_i)\big),$$

therefore by the triangle inequality, Proposition B.5, and Lemma B.6,

$$(c) \leq \frac{1}{i} \sum_{r \leq i} \|z_r - \bar{v}_r\| + \frac{1}{i} \sum_{r \leq i} \|\bar{v}_r - \bar{v}_i\| \leq B^3 \sqrt{H} \big(\Delta_h^{(l)}\big)^2 + \frac{1}{i} \sum_{r \leq i} \max_{k \leq i} \|v_k - \bar{v}_i\|$$

$$= B^3 \sqrt{H} \big(\Delta_h^{(l)}\big)^2 + \max_{k \leq i} \|v_k - \bar{v}_i\|$$

$$\leq B^3 \sqrt{H} \big(\Delta_h^{(l)}\big)^2 + B\Delta_h^{(l)}$$

To finish the proof, take the maximum over $i$ and $j \leq i$. $\square$

**Proposition B.8** (Full Transformer block recursion)**.** *There exist constants $A_1, A_2$ depending only on $B$, and $H$, such that*

$$\Delta_h^{(l+1)} \leq A_1 \Delta_h^{(l)} + A_2 \big(\Delta_h^{(l)}\big)^2.$$

*Proof.* From Proposition B.7, the single-head spread is bounded by a linear term $2B\Delta_h$ plus a quadratic term $2B^3\sqrt{H}$. Concatenation and $W_O$ multiply by at most $\|W_O\|$ (up to a fixed constant depending on number of heads). Adding the residual preserves a linear contribution in $\Delta_h^{(\ell)}$. The positionwise LayerNorm/MLP, being $B$-Lipschitz, scales the spread by at most $B$. Collecting the constants into $A_1$ and, $A_2$ gives the desired result. □

We can now proof the full propagation result.

**Theorem B.9.** *For any finite depth L, there exists $\varepsilon > 0$ (depending on B, L, and H) such that if $\Delta_h^{(1)} \le \varepsilon$, then for all $l \le L$,*

$$\Delta_h^{(l)} \le C\Delta_h^{(1)} \le C\varepsilon,$$

*with $C = C(B, L, H)$.*

*Proof.* By Proposition B.8, $\Delta_h^{(l+1)} \le A_1\Delta_h^{(l)} + A_2(\Delta_h^{(l)})^2$. Choose $\varepsilon \le \min\{1, (A_1/A_2)\}$ so that $A_2\Delta_h^{(l)} \le A_1$. Then $\Delta_h^{(l+1)} \le 2A_1\Delta_h^{(l)}$. Induction yields $\Delta_h^{(l)} \le (2A_1)^{l-1}\Delta_h^{(1)} \le C\Delta_h^{(1)}$ for $l \le L$ with $C = (2A_1)^{L-1}$. □

This conclude the first part of the proof, regarding uniformity propagation across depth. Note that the bounds in the proof *do not* depend on the number of tokens in the input sequence.

**$\mathbf{A}^c$ bound.** Recall that,

$$\mathbf{A}^c = \frac{1}{T}\sum_{i=1}^{T}\sum_{j\le i}\alpha_{ij}c_{ij}$$

where $c_{ij}$ are centered positional weights, i.e. $\sum_{j\le i}c_{ij} = 0$. For any such $c_{ij}$ we have

$$\left|\mathbf{A}^c\right| = \frac{1}{T}\left|\sum_{i=1}^{T}\sum_{j\le i}\alpha_{ij}c_{ij}\right| = \frac{1}{T}\left|\sum_{i=1}^{T}\sum_{j\le i}\alpha_{ij}c_{ij}\right| = \frac{1}{T}\left|\sum_{i=1}^{T}\sum_{j\le i}(\alpha_{ij}-\tfrac{1}{i})c_{ij} + \sum_{i=1}^{T}\underbrace{\sum_{j\le i}\tfrac{1}{i}c_{ij}}_{=0}\right|$$

$$\le \frac{1}{T}\sum_{i=1}^{T}\sum_{j\le i}|\alpha_{ij}-\tfrac{1}{i}||c_{ij}| \le \Big(\underbrace{\max_{1\le j\le i\le T}|c_{ij}|}_{C}\Big)\frac{1}{T}\sum_{i=1}^{T}\|\alpha_i - u_i\|_1$$

$$\le CB^2\sqrt{H}\Delta_h^{(l)} = \mathcal{O}(\varepsilon).$$

**Q/K gradient bounds.** Let $g_{ij} = \partial\mathbf{A}^c/\partial s_{ij}$. We have

$$g_{ij} = \tfrac{1}{T}\alpha_{ij}(c_{ij} - c_i^{\alpha}),$$

where $c_i^{\alpha} = \sum_{p\le i}\alpha_{ip}c_{ip}$.

**Lemma B.10.** *For every i, $\sum_{j\le i}g_{ij} = 0$, and therefor for any vectors $a_j$*

$$\sum_{j\le i}g_{ij}a_j = \sum_{j\le i}g_{ij}(a_j - \bar{a}_i).$$

*Proof.* First notice that

$$\sum_{j\le i}g_{ij} = \frac{1}{T}\sum_{j\le i}\alpha_{ij}(c_{ij} - c_{ij}^{\alpha}) = \frac{1}{T}\mathbb{E}_{j\sim\alpha_i}[c_{ij} - \mathbb{E}_{p\sim\alpha_i}[c_{ip}]] = 0.$$

For the second part, observe that

$$\sum_{j\le i}g_{ij}(a_j - \bar{a}_i) = \sum_{j\le i}g_{ij}a_j - \bar{a}_i\sum_{j\le i}g_{ij} = \sum_{j\le i}g_{ij}a_j.$$

□

Now, from direct computation and an application of Lemma B.10, we have

$$\frac{\partial \mathbf{A}^c}{\partial W_Q} = \frac{1}{\sqrt{d_k}} \sum_{i=1}^T \Big( \sum_{j \leq i} g_{ij} k_j \Big) h_i^\top = \frac{1}{\sqrt{d_k}} \sum_{i=1}^T \sum_{j \leq i} g_{ij} (k_j - \bar{k}_i) h_i^\top,$$

$$\frac{\partial \mathbf{A}^c}{\partial W_K} = \frac{1}{\sqrt{d_k}} \sum_{i=1}^T q_i \Big( \sum_{j \leq i} g_{ij} h_j^\top \Big) = \frac{1}{\sqrt{d_k}} \sum_{i=1}^T \sum_{j \leq i} g_{ij} q_i (h_j - \bar{h}_i)^\top.$$

Let's analyse the norm:

$$\Big\| \frac{\partial \mathbf{A}}{\partial W_K} \Big\| = \Big\| \frac{1}{\sqrt{d_k}} \sum_{i=1}^T \sum_{j \leq i} g_{ij} q_i (h_j - \bar{h}_i)^\top \Big\| \leq \frac{1}{\sqrt{d_k}} \sum_{i=1}^T \sum_{j \leq i} |g_{ij}| \, \|q_i\| \, \|h_j - \bar{h}_i\|$$

$$\leq B\sqrt{H} \Delta_h^{(l)} \sum_{i=1}^T \sum_{j \leq i} |g_{ij}|$$

$$\leq \frac{B\sqrt{H} \Delta_h^{(l)}}{T} \Big( \sum_{i=1}^T \sum_{j \leq i} |(\alpha_{ij} - \tfrac{1}{i})(c_{ij} - c_i^\alpha)| + \sum_{i=1}^T \sum_{j \leq i} \tfrac{1}{i} |c_{ij} - c_i^\alpha| \Big)$$

$$\leq B\sqrt{H} \Delta_h^{(l)} \Big( B^2 \sqrt{H} \Delta_h^{(l)} C + C \Big) = \mathcal{O}(\varepsilon),$$

where $C = \max_{1 \leq j \leq i \leq T} |c_{ij} - c_i^\alpha| \leq \max_{1 \leq j \leq i \leq T} |c_{ij}|$. An analogous result holds for $W_Q$,

$$\Big\| \frac{\partial \mathbf{A}}{\partial W_Q} \Big\| = \Big\| \frac{1}{\sqrt{d_k}} \sum_{i=1}^T \sum_{j \leq i} g_{ij} (k_j - \bar{k}_i) h_i^\top \Big\| \leq \frac{1}{\sqrt{d_k}} \sum_{i=1}^T \sum_{j \leq i} |g_{ij}| \, \|k_j - \bar{k}_i\| \, \|h_i\|$$

$$\leq B\sqrt{H} \Delta_h^{(l)} \sum_{1 \leq j \leq i \leq T} |g_{ij}|$$

$$\leq \frac{B\sqrt{H} \Delta_h^{(l)}}{T} \Big( \sum_{i=1}^T \sum_{j \leq i} |(\alpha_{ij} - \tfrac{1}{i})(c_{ij} - c_i^\alpha)| + \sum_{i=1}^T \sum_{j \leq i} \tfrac{1}{i} |c_{ij} - c_i^\alpha| \Big)$$

$$\leq B\sqrt{H} \Delta_h^{(l)} \Big( B^2 \sqrt{H} \Delta_h^{(l)} C + C \Big) = \mathcal{O}(\varepsilon).$$

This concludes the proof of Theorem 3.4.

## C  EXPERIMENTAL DETAILS

### C.1  TRAINING

**DroPE from a RoPE transformer trained from scratch.** For the first part of our experimental evaluation, we train a small RoPE transformer with almost half a billion parameters on FineWeb (Penedo et al., 2024) for over 16B tokens with a sequence length of 1024. We note this is well over 2 times the chinchilla optimal number of tokens from Hoffmann et al. (2022). We use a QWEN2 (Yang et al., 2024) tokenizer and follow the specifications (number of layers/hidden dimensions) from the 0.5B model from the same family. We implemented all our baselines on top of this architecture, pretraining them for the same large number of tokens. We use the AdamW optimizer Loshchilov & Hutter (2017) with a small warmup phase of 520 steps, a batch size of 1024, a peak learning rate of $3.0 \times 10^{-4}$, and a cosine decay thereafter. For DroPE we followed a similar optimization setup, but only training for 2B total tokens using a shorter warmup of 70 steps and a slightly larger learning rate of $1.0 \times 10^{-3}$ to compensate for the shorter training budget. We provide a full list of hyperparameters and training specifications for this setting in the left column of Table 5.

**DroPE from a pretrained SMOLLM .** For the second part of our experimental evaluation, we use a SMOLLM (Allal et al., 2024) with around 362 million parameters already extensively pretrained on the SmolLM corpus (Ben Allal et al., 2024) for over 600B tokens with a sequence length of 2048 – almost 100 times the chinchilla optimal number. This model used a GPT2 (Radford et al., 2019)

| Pretraining and DroPE Hyperparameter | RoPE transformer | SMOLLM |
|---|---|---|
| **Model architectures** | | |
| Model parameters | 494M | 362M |
| Model parameters w/o embeddings | 358M | 315M |
| Hidden size | 896 | 960 |
| Hidden MLP size | 4864 | 2560 |
| Hidden activation | SiLU | SiLU |
| Number of hidden layers | 24 | 32 |
| Number of attention heads | 14 | 15 |
| Number of key–value heads | 2 | 5 |
| Head dimension | 64 | 64 |
| Attention bias | false | false |
| Attention dropout | 0.0 | 0.0 |
| Initializer range | 0.02 | 0.02 |
| RoPE $\theta$ | 1,000,000 | 10,000 |
| Tied word embeddings | true | true |
| Output router logits | true | true |
| Computation dtype | bfloat16 | bfloat16 |
| Tokenizer | QWEN2 | GPT2 |
| **Pretraining setup** | | |
| Optimizer | AdamW | AdamW |
| Learning rate | $3.0 \times 10^{-4}$ | $3 \times 10^{-3}$ |
| Weight decay | 0.1 | 0.1 |
| Adam parameters $(\beta_1, \beta_2, \epsilon)$ | $(0.9, 0.95, 1 \times 10^{-8})$ | $(0.9, 0.95, 1 \times 10^{-8})$ |
| Learning rate scheduler | Cosine decay | Cosine decay |
| Warmup steps | 520 | N/A |
| Maximum sequence length | 1024 | 2048 |
| Global train batch size (sequences) | 1024 | 512 |
| Tokens per training step | 1,048,576 | 1,048,576 |
| Total tokens | 16.8B | 600B |
| Dataset | fineweb | smollm-corpus |
| **DroPE setup** | | |
| QK-norm | False | True |
| Optimizer | AdamW | AdamW |
| Learning rate | $1.0 \times 10^{-3}$ | $1.0 \times 10^{-3}$ |
| Weight decay | 0.1 | 0.1 |
| Adam parameters $(\beta_1, \beta_2, \epsilon)$ | $(0.9, 0.95, 1 \times 10^{-8})$ | $(0.9, 0.95, 1 \times 10^{-8})$ |
| Learning rate scheduler | Cosine decay | Cosine decay |
| Warmup steps | 70 | 490 |
| Maximum sequence length | 1024 | 2048 |
| Global train batch size (sequences) | 1024 | 512 |
| Tokens per training step | 1,048,576 | 1,048,576 |
| Total tokens | 2.10B | 31.46B/62.9B/125.8B |
| Dataset | fineweb | fineweb-edu |

Table 5: Architectures, optimization, and other training setup hyperparameters for pretraining our RoPE transformer, SMOLLM, and our two new DroPE phases.

tokenizer and its architecture was designed to be similar to models of the LLAMA2 family (Touvron et al., 2023). While not all training details have been disclosed, Allal et al. (2024) explicitly mentions using the AdamW optimizer Loshchilov & Hutter (2017), a batch size of 512, a peak learning rate of $3.0 \times 10^{-3}$, and a cosine decay thereafter. For DroPE we again tried to follow a similar optimization setup, across our different 30B/60B/120B training regimes, introducing a short warmup of 490 steps and a slightly lower learning rate of $1.0 \times 10^{-3}$ as we found their reported $3.0 \times 10^{-3}$ led to instabilities from the small batch size. Given the more extended training period, we used a simple QKNorm (Henry et al., 2020) after dropping the positional embeddings, which we found beneficial to mitigate sporadic instabilities from large gradients. We note that preliminary experiments showed that normalizing only the queries led to even faster learning and also successfully stabilized long

training. We believe further exploration of this new Q-norm method could be an exciting direction for future work to train transformers without positional embeddings at even larger scales. We provide a full list of hyperparameters and training specifications for this setting in the right column of Table 5.

## C.2 EVALUATION

**Needle-in-a-haystack.** We evaluate long-context retrieval using the *needle-in-a-haystack* (NIAH) setup, which places a short "needle" inside a long distractor "haystack." Following prior work (Kamradt, 2023), our haystack is a random excerpt from Paul Graham's essays, and each needle is a seven-digit "magic number" paired with a short key/descriptor. We study three variants:

- **(Standard NIAH)** We insert a single needle and prompt the model to retrieve it.
- **Multi-Query NIAH:** We insert multiple (key, value) pairs and prompt the model to return as many values as possible for a given list of keys. For example: `The special magic numbers for whispering-workhorse and elite-butterfly mentioned in the provided text are:`.
- **(Multi-Key NIAH)** We insert multiple (key, value) pairs but query for a single key, e.g., `The special magic number for elite-butterfly mentioned in the provided text is:`
- **(Multi-Value NIAH)** We associate multiple values with one key and ask for all of them without pointing to specific positions, e.g., `What are all the special magic numbers for cloistered-colonization mentioned in the provided text?`

Inserted needles and example targets are formatted in natural language, e.g., `One of the special magic numbers for whispering-workhorse is: 1019173` and `One of the special magic numbers for elite-butterfly is: 4132801`. For the standard NIAH variant, we report the average success rate over all possible needle depths. For the multiple needles NIAH variants, we always insert four (key, value) needle pairs, placed at random sequence locations. Unless otherwise noted, we use greedy decoding (logit temperature $= 0$) for reproducibility.

**Long-context evaluations.** We use standard implementations of PI, RoPE-NTK, and YaRN. For tasks that require a fixed maximum context length (e.g., NIAH at $2\times$ the training context), we set the *extension factor $s$* manually. For settings that require reasoning across multiple context lengths and extended generations, we employ a *dynamic scaling* schedule that adjusts $\gamma$ as a function of the generation length as detailed in Peng et al. (2023).

For DroPE, we follow Wang et al. (2024) and apply softmax *temperature scaling* when evaluating on longer sequences. In practice, we tune a single scalar logit scale (equivalently, the inverse temperature) on a held-out set at the target length. Analogous to (Peng et al., 2023), we fit this coefficient by minimizing perplexity to obtain the optimal scaling. For the DroPE model trained from scratch, the best-performing scale is

$$\beta^{\star} = 1 + 0.412 \ln(s),$$

and for SMOLLM-DROPE the optimal scale is

$$\beta^{\star} = 1 + 0.103 \ln(s),$$

Where $s = C_{\text{test}}/C_{\text{train}}$ is the context extension factor. Unless otherwise specified, all other decoding settings are held fixed across lengths.

**Language modeling benchmarks.** We evaluate SMOLLM and SMOLLM-DROPE on six standard multiple-choice benchmarks using the LIGHTEVAL harness (Habib et al., 2023): **ARC-E/C:** grade-school science QA split into Easy and Challenge sets, the latter defined by questions that defeat simple IR and co-occurrence baselines (Clark et al., 2018); **HellaSwag:** adversarially filtered commonsense sentence completion that is easy for humans but challenging for LMs (Zellers et al., 2019); **OpenBookQA:** combining a small "open book" of science facts with broad commonsense to answer 6K questions (Mihaylov et al., 2018); **PIQA:** two-choice physical commonsense

Table 6: **DroPE matches base model in-context performance.** Comparison of the pretrained SMOLLM model with SMOLLM-DROPE evaluated on variety of LM benchmarks.

| Model | ARC-E | ARC-C | HellaSwag | OpenBookQA | PIQA | Winogrande | Avg. |
|---|---|---|---|---|---|---|---|
| SmolLM | 65.6 | 36.0 | 53.8 | 37.2 | **72.0** | **53.7** | 53.1 |
| SmolLM-DroPE | **67.3** | **37.6** | **53.9** | **38.0** | 71.5 | 52.3 | **53.4** |

Table 7: Validation perplexity for a 500M-parameter transformer trained on 16B tokens, when dropping positional encodings at different stages of pretraining.

| | DroPE @ 0K (NoPE) | DroPE @ 8K | DroPE @ 14K | DroPE @ 16K (RoPE) |
|---|---|---|---|---|
| Validation perplexity | 23.77 | 22.42 | 21.73 | **21.72** |

reasoning (Bisk et al., 2020); and **WinoGrande:** a large-scale, adversarial Winograd-style coreference/commonsense benchmark (Sakaguchi et al., 2021). We follow the harness defaults for prompt formatting, decoding, and scoring, and do not perform any task-specific fine-tuning or data adaptation.

# D  ADDITIONAL EXPERIMENTAL RESULTS

**When should we start recalibration?**  In this setup, we train a 500M-parameter transformer on 16B tokens and remove its PEs during training. We vary the training step at which recalibration is activated. We consider four recipes:

- Dropping PEs from step 0 (*NoPE transformer*),
- Dropping PEs at step 8K,
- Dropping PEs at step 14K,
- Dropping PEs at step 16K (*RoPE transformer*, i.e., no dropping during training).

Table 7 reports the final validation perplexity for each setting.

We find that this ablation further strengthens our theoretical observation that DroPE should be integrated later in training. Our analysis in Section 3 suggests that NoPE transformers struggle to train efficiently, whereas retaining RoPE for most of training benefits optimization. Consistent with this, we observe that dropping the positional encoding only at the very end of pretraining (DroPE @ 16K) yields the best validation perplexity, while earlier dropping steadily degrades performance.

Finally, we emphasize that in this setup DroPE does not incur additional training cost: the total number of optimization steps is unchanged, and once the positional encoding is removed, training becomes slightly faster due to skipping the RoPE rotation operations in attention.

**Average LongBench scores and tasks breakdowns.**

Table 8: Average performance over all LongBench tasks for different RoPE scaling methods.

| Method | Avg. LongBench score |
|---|---|
| SMOLLM | 2.59 |
| SMOLLM + PI | 2.48 |
| SMOLLM + RoPE-NTK | 12.21 |
| SMOLLM + YaRN | 13.07 |
| SMOLLM-DROPE | **13.81** |

**Effect of QKNorm and loss spike.** Here we clarify the role of QKNorm Henry et al. (2020) in our architecture and verify that it is not a confounding factor for the effectiveness of DroPE. QKNorm was introduced strictly as an optimization-stability mechanism to enable training with higher learning rates, following recent practices in large-scale model training such as OLMo2 (OLMo et al.,

Table 9: MultiFieldQA performance across context length buckets for SMOLLM variants.

| Model | 0–4K (0–2× ctx) | 4–8K (2–4× ctx) | 8–16K (4–8× ctx) |
|---|---|---|---|
| SmolLM-DroPE | 32.82 | 24.73 | 30.07 |
| SmolLM-NTK | 34.25 | 22.30 | 21.63 |
| SmolLM-YaRN | 33.96 | 22.91 | 20.08 |

Table 10: MuSiQue performance across context length buckets for SMOLLM variants.

| Model | 0–4K (0–2× ctx) | 4–8K (2–4× ctx) | 8–16K (4–8× ctx) | 16–32K (8–16× ctx) |
|---|---|---|---|---|
| SmolLM-DroPE | 50.00 | 6.11 | 8.05 | 16.67 |
| SmolLM-NTK | 0.00 | 4.36 | 3.36 | 0.00 |
| SmolLM-YaRN | 0.00 | 19.68 | 3.13 | 7.14 |

2024) and Qwen3 (Yang et al., 2025a), where normalization is used primarily to stabilize gradients and mitigate loss spikes rather than to increase modeling capacity.

To assess the interaction between QK Norm and DroPE, we conducted a controlled ablation study on the SmolLM-360M model using six configurations: three learning rates ($3 \times 10^{-5}$, $3 \times 10^{-4}$, $10^{-3}$), each trained with and without QK Norm.

The results, summarized in Table 11, yield two main observations:

- **Lower learning rates** ($3 \times 10^{-5}$, $3 \times 10^{-4}$). DroPE works effectively without QKNorm. At the lowest learning rate ($3 \times 10^{-5}$), the model without QK Norm achieves a slightly better final loss (2.713 vs. 3.102). Together with the $3 \times 10^{-4}$ setting (2.530 vs.2.555), this indicates that QK Norm does not consistently improve performance in low-volatility regimes and is not the source of our gains.
- **High learning rate** ($10^{-3}$). At the highest learning rate, the model without QKNorm becomes unstable (loss spikes, gradient explosions), leading to poor convergence (final loss 6.334). In contrast, adding QKNorm stabilizes training and allows us to leverage the higher learning rate to achieve the best overall performance (final loss 2.496).

Figure 12 shows the corresponding training curves with and without QK Norm, highlighting the presence of loss spikes at higher learning rates, in line with observations reported in **?**. These results empirically demonstrate that the primary role of QK Norm is to act as a stabilizer that enables the use of a more aggressive, compute-efficient learning rate. Importantly, DroPE can still be applied without QK Norm by using a moderate learning rate (e.g., ($3 \times 10^{-4}$), which is our default setting for all experiments except the longer SmolLM-360M recalibration phases.

Table 11: Ablation study on SmolLM-360M recalibration with and without QK Norm across different learning rates.

| Learning Rate | With QK Norm | Without QK Norm | Status |
|---|---|---|---|
| $10^{-3}$ (High) | 2.496 | 6.334 | Unstable without Norm |
| $3 \times 10^{-4}$ (Mid) | 2.555 | 2.530 | Stable / Comparable |
| $3 \times 10^{-5}$ (Low) | 3.102 | 2.713 | Stable / Comparable |

# E  LARGE LANGUAGE MODEL (LLM) USAGE

We used large language models to assist with refining the wording in some sections of this paper. Their role was limited to improving clarity of technical explanations, grammar, style, and overall readability. All research ideas, methodology, experiments, analysis, and conclusions are entirely our own; the models were employed solely as writing aids.

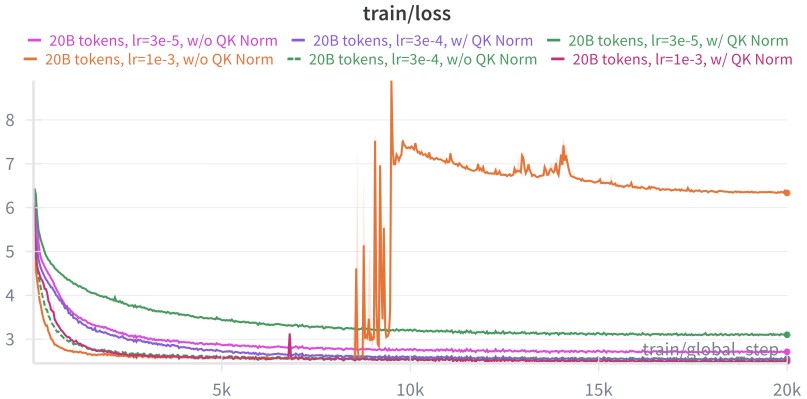

Figure 12: QKNorm allows for recalibration at a higher learning rate.

