# OpenReview forum: "Extending the Context of Pretrained LLMs by Dropping Their Positional Embedding"
_ICLR.cc/2026/Conference — ICLR 2026 Poster_

### Official Review · Reviewer_fTZx · 2025-10-25

**Soundness:** 3
**Presentation:** 2
**Contribution:** 4
**Rating:** 6
**Confidence:** 4

**Summary:**

The paper challenges the conventional methodology by bravely removing positional encoding at serving time (NoPE), but the model had been initially trained using RoPE and recalibrated with continuous pretraining in a later phase without positional encoding.

For the background, it is well-known that NoPE
1. can learn positions of tokens due to causal mechanisms in attention;
2. but underperforms in terms of negative log-likelihood when comparing with most existing positional encodings.
Although NoPE is known to have good generalization properties in terms of context length, it is generally dismissed as the widespreading intuition is that the positional information is "hard to learn" or even "lost".

The paper provides an explanation of why NoPE has difficulties in learning positional information. On the other hand, it also provides some observations to explain why RoPE, though learning the positional information fast, has trouble with length generalizations.

**Strengths:**

1. Simplicity.
2. The idea is fairly novel.
3. Results are fairly compelling with good theoretic analysis to back them up.
4. Potential impact. I personally think this inspires the search of better positional encoding by not only looking at architecture manipulations, but also the training process.

**Weaknesses:**

- One of the biggest concern to widely adopt this in LLM training is a rigorous study to confirm potential risks. There are two aspects that I'm particular concerned, despite the good numbers on academic benchmarks
  1. The loss spike and cooling down, does it hurt further learning dynamics? What does the spike do to other layers' parameters, parameter/grad norms, etc. Would it make further SFT different?
  2. This is less important as to a "weakness", but how does it fare on other more up-to-date LLM evaluations? This might be relevant if we want to understand if it is genuinely without regression on all tasks, or whether there are some issues that we need to understand.
- There seems to be a lack of experimental rigorousness. For this I still very much want to understand all these comparison setups, what are held as control variables and how the confounders are ruled out. Also there seems to be a sneaky modification of architecture in Appendix C.1, even though they "look minor", introducing unwanted variable makes experiments shaky. See my questions below.

**Questions:**

It's possible that I miss certain things already in the appendix, but below are the questions I want to understand better.

- I don't understand Figure 9 and Table 4: are you comparing the original SmolLM with SmolLM + extra 30/60/120B token's finetuning? Or did you match an equi-token setup and further finetune SmolLM for the same amount of tokens? If the former, it is perhaps not entirely fair and it still bears the question as to how much the original model would trend higher given the additional training. Also, do you have certain measurement of confidence interval on those eval accuracies?
- In Table 2, what are the exact setup for SmolLM+PI/NTK/YaRN, and which token budget for the DroPE did you use?
- In Appendix C.1, you mentioned a "simple QKNorm". Does that mean you modified the architecture slightly after dropping the positional encoding? What happened without? Are we able to see a detailed analysis on ablating all these variables?

I will read more later and write more questions if something arises.

I am tentatively giving it a 6 as I think it is very intriguing work and the authors have very good math foundations and taste. But experimental rigorousness is crucial. **I may revise the score higher or even lower** as I understand more about the details of the work.

---

> ### Author Response · Authors · 2025-11-26
> **Authors' Response to Reviewer fTZx (Part 1/5)**
>
> We thank the reviewer for a careful reading of the paper, for their insightful comments, and for noting our paper's **simplicity**, **novelty**, **compelling results**, **theoretical analysis**, and **impact potential**. We also appreciate the reviewer's concerns and questions and address them in detail below:
>
> **Weaknesses**
> > 1. The loss spike and cooling down, does it hurt further learning dynamics? What does the spike do to other layers' parameters, parameter/grad norms, etc. Would it make further SFT different?
>
> This is a great question! To answer it, we look at the gradient norm before, after, and during the application of DroPE in the training of our 500M parameter model, illustrated in Figure 2. Before the application of DroPE (first 14K steps) is 0.18, immediately after applying DroPE (the next 500 steps), the average gradient norm jumps to 8.19, but it drops back to 0.15 for the rest of the training, similar to the full RoPE training run. Additionally, to test the empirical effect of DroPE on SFT performance, we also run new experiments finetuning our SmolLM-DroPE model on the smoltalk dataset [6]. After one epoch, our model achieves a mean-token accuracy of **82.32%**, matching the performance of the base model finetuned on the same dataset. In fact, we posit that instruction tuning is even less sensitive to PE due to the added positional signal coming from the chat template, and the fact that after the pre-training, the model is only concerned with the restricted conditional generation task.
>
> Following the reviewer's feedback, we added our gradient norm graphs and preliminary SFT results to the latest version of our manuscript, in order to reinforce the evidence for our method's robustness.
>
> > 2. This is less important as to a "weakness", but how does it fare on other more up-to-date LLM evaluations? This might be relevant if we want to understand if it is genuinely without regression on all tasks, or whether there are some issues that we need to understand.
>
> Following the reviewer feedback, we extended our evaluation by **adding new experiments and analysis** to Section 5 of the updated manuscript, with a larger **variety of base models** at even **larger scales**, and more **comprehensive evaluation** considering and analyzing different extrapolation distances and more challenging tasks. We hope these results will provide further validation of DroPE's robustness to a variety of base models and evaluation settings.
>
> **Part 1: Base models tested.** In Section 5, we have added new experiments to the latest revision, considering **two additional pretrained LLMs** with increasing numbers of parameters and pretraining context lengths.
>
> First, we considered a larger 1.7B model from the same SmolLM family we originally used. This larger model was also pretrained with a higher budget of 1T tokens. We applied DroPE and then recalibrated this model on only 20B tokens, 2% of the original training budget. As shown in the table below, despite the much lower recalibration budget in relative terms, our findings for this larger model are consistent with the ones obtained using the 360M variant, with DroPE outperforming other context extension techniques.
>
>   | Dataset       | Base  | NTK   | YaRN  |   PI  |   DroPE   |
>   |---------------|-------|-------|-------|-------|-----------|
>   | MultifieldQA  | 4.12  | 27.58 | 27.60 | 4.12  | **32.18** |
>   | MusiQue       | 0.50  | 3.37  | 3.90  | 0.50  | **7.53**  |
>   | GovReport     | 4.70  | 24.65 | 17.19 | 4.70  | **24.77** |
>
> Second, we considered Llama2-7B [2], originally pretrained on 4T tokens with 4096 context length, recalibrated once again for 20B tokens, this time only 0.5% of the original budget. We chose this model as the Llama family was suggested in the reviewer's question 3 below. We note that due to model scale and the increased context, merely adapting this model for 20B tokens already took more than 2000 H100 GPU hours. Furthermore, as the team behind Llama2-7B did not disclose the details of their pretraining data, we still resorted to the SmolLM training corpus. Nonetheless, despite having had to push the recalibration phase budget to the minimum in relative terms and not having access to the Llama closed-source corpus, we found our Llama7B-DroPE model still outperforms all RoPE-context extension scaling baselines on LongBench.
>
>   | Dataset      | Base  | NTK   | YaRN  | DroPE |
>   |-------------|-------|-------|-------|-------|
>   | MultifieldQA| 17.26 | 21.81 | 23.13 | **25.90** |
>   | MusiQue     | 10.43 | 10.91 | 7.65  | **12.88** |
>   | GovReport   | 32.41 | 32.91 | 26.65 | **39.47** |

---

> > ### Author Response · Authors · 2025-11-26
> > **Authors' Response to Reviewer fTZx (Part 2/5)**
> >
> > **Part 2: Additional benchmarks and challenging tasks.** Our original evaluation considered seven pretraining tasks [1] for in-context evaluation, three synthetic tasks from the RULER benchmark [10] where we chose to evaluate at 2x our model's training context length, and five challenging LongBench tasks with up to 32K tokens corresponding to 16x of our model's original context length. We have now extended our evaluation settings with a detailed analysis of how varying the context extension lengths affects performance, to check for potential failure cases beyond 2x, and evaluated our method on additional tasks and benchmarks.
> >
> > First, to analyze how varying context length affects performance at a granular level beyond 2x, we started by evaluating our models on NIAH tasks of up to 8x of our original context length:
> >
> >   | Context length            |   DroPE   |  NTK  | YaRN  |
> >   |---------------------------|-----------|-------|-------|
> >   | 4K (2x original ctx)      | **74.92** | 29.84 | 48.25 |
> >   | 8K (4x original ctx)      | **55.00** | 14.37 | 25.62 |
> >   | 16K (8x original ctx)     | **52.50** |  7.19 | 12.18 |
> >
> > Additionally, we also recorded granular performances by considering the scores attained by the different context extension strategies, by partitioning LongBench problems by their input context lengths. To remove confounding factors, we focused this analysis on two tasks that include problems across all our considered context length evaluation brackets up to x8 and x16:
> >
> > **Multifield QA**
> > | model        | 0-4K (0-2x ctx) | 4-8K (2-4x ctx) | 8-16K (4-8x ctx) |
> > | ------------ | --------------- | --------------- | ---------------- |
> > | SmolLM-DroPE | 32.82           | **24.73**       | **30.07**        |
> > | SmolLM-NTK   | **34.25**       | 22.3            | 21.63            |
> > | SmolLM-YaRN  | 33.96           | 22.91           | 20.08            |
> >
> > **MusiQue**
> >
> > | model        | 0-4K (0-2x ctx) | 4-8K (2-4x ctx) | 8-16K (4-8x ctx) | 16-32K (8-16x ctx) |
> > | ------------ | --------------- | --------------- | ---------------- | ------------------ |
> > | SmolLM-DroPE | **50**          | 6.11            | **8.05**         | **16.67**          |
> > | SmolLM-NTK   | 0               | 4.36            | 3.36             | 0.0                |
> > | SmolLM-YaRN  | 0.0             | **19.68**       | 3.13             | 7.14               |
> >
> > As shown by our new analysis results above, while there is a drop in performance from 2x to 8x (74.92% to 52.50%),  our new method still significantly outperforms other RoPE-extension baselines across all extension factors. Interestingly, these results also reveal that beyond a 4x context extension factor, DroPE's performance appears to stabilize while the other methods sharply deteriorate. This is most evident on the synthetic NIAH tasks, where we are able to precisely control the distance to the information to retrieve for success. In this instance, we see DroPE's performance drops by only 2.5% (from 55% to 52.5%) when doubling the context length from 4x to 8x. In contrast, when applying NTK and YaRN, model performance halves (14.37% to 7.19% for NTK, and 25.62% to 12.18% for YaRN).
> >
> > Second, to expand the comprehensiveness of our evaluation, we started by expanding our LongBench evaluation to the full benchmark of 26 tasks. We note that in our original work, we chose to focus on five particular tasks as we wanted to minimize confounding factors due to the noisiness and difficulty of some tasks in the LongBench benchmark [4], where even powerful closed-source GPT models achieve less than 5% (e.g, Llama3 [11] also only evaluates on 2 tasks from the suite)
> >
> > | Dataset                      | Base | PI   | NTK   | YaRN  | DroPE |
> > |------------------------------|------|------|-------|-------|-------|
> > | Average (all LongBench tasks)| 2.59 | 2.48 | 12.21 | 13.07 | **13.81** |
> >
> > Moreover, we also considered a new long document evaluation setup, running our model on the full NarrativeQA [9] in its original settings without filtering. We evaluated using Llama2-7B as the base model and compared it to both YaRN and our DroPE version. Results are reported as ROUGE-L scores.
> >
> > |Base|YaRN|DroPE|
> > |-|-|-|
> > |0.0|6.98|**10.93**|
> >
> > In general, we find that DroPE excels at tasks that require reasoning or analysis on multiple pieces of data that appear deep in the context. These types of tasks require strong recall capabilities for which semantic entropy heads, that are preserved by DroPE, are particularly useful.
> >
> > By expanding the set of base models, the analysis, and the benchmark breadth, we hope these new results will reinforce the evidence of the generality and help address the reviewer's feedback about testing DroPE in a wider range of evaluation settings.

---

> > > ### Author Response · Authors · 2025-11-26
> > > **Authors' Response to Reviewer fTZx (Part 3/5)**
> > >
> > > > 3. There seems to be a lack of experimental rigorousness. For this I still very much want to understand all these comparison setups, what are held as control variables and how the confounders are ruled out. Also there seems to be a sneaky modification of architecture in Appendix C.1, even though they "look minor", introducing unwanted variable makes experiments shaky. See my questions below.
> > >
> > > In Section 5 of our initial submission, we run experiments considering two main settings using different model scales:
> > > - We used a 500M parameter RoPE transformer trained to $2\times$ the chinchilla optimal rate to show that DroPE can be integrated as a *zero cost* extension at the end of standard pretraining pipelines, in order to perform targeted controlled experiments and compare DroPE with prior baselines.
> > > - We used the SmolLM-360M model pretrained on 600B tokens to show that DroPE can be applied to *adapt LLMs in the wild* at a fraction of their original training budget and study how far this can be pushed in terms of sample efficiency.
> > >
> > > **Questions**
> > > > 1. I don't understand Figure 9 and Table 4: are you comparing the original SmolLM with SmolLM + extra 30/60/120B token's finetuning? Or did you match an equi-token setup and further finetune SmolLM for the same amount of tokens? If the former, it is perhaps not entirely fair and it still bears the question as to how much the original model would trend higher given the additional training. Also, do you have certain measurement of confidence interval on those eval accuracies?
> > >
> > > As described in our response to Weakness 3, we run experiments considering two main settings: a 500M parameter RoPE transformer to show that DroPE can be integrated as a *zero cost* extension at the end of standard pretraining pipelines (i.e, equating the total amount of RoPE + DroPE tokens), and our SmolLM experiments where we showed that DroPE can be applied to *adapt LLMs in the wild* at a fraction of their original training budget. Thus, the reviewer's understanding is indeed correct: we are comparing SmolLM to SmolLM-DroPE after a recalibration, performed by "continued pretraining" on 30, 60, and 120B tokens. The goal of this experiment is to demonstrate DroPE's effectiveness in cases where we don't have access to the model's training pipeline. This is in contrast to the previous set of experiments, where, since we consider a controlled setup with a language model that we train from scratch, we are able to restart training from intermediate checkpoints. However, we would like to note a few aspects, for which we believe the additional tokens do not compromise the soundness of this evaluation:
> > > - According the the SmolLM blogpost [1], the additional tokens we train on (taken from FineWeb-Edu [4]) are part the SmolLM's pretraining dataset. Previous research shows that retraining on the same token has a negligible effect [2].
> > > - The goal of our recalibration is _not_ to improve on the in-context performance of the base model but rather to make the model generalize better to longer sequences. The goal of this set of experiments is to demonstrate that even with a base model that was trained on a large corpus, the post-training performance can be recovered by recalibration.
> > > - As seen in Figure 9, the **vast majority** of the original performance is restored after only 5B tokens (< 1% of original training budget), the rest of the recalibration is meant to demonstrate that the full original performance can, in fact, be restored. This is also confirmed with our newly added experiments on Llama2 and SmolLM model-1.7B (see response to W2), where we only train for 20B tokens (0.5% of 2% of the training budget), an even more negligible amount compared to the pretraining cost.
> > >
> > > The only implementation difference between these two experiments is the addition of QK norm for the Smollm-360M model, whose benefit comes from allowing us to train with higher learning rates without suffering training spikes common in extended LM pretraining (following the reviewer's feedback, we added a detailed ablation of this in Appendix C, see our response to Q3).
> > >
> > > We note that all our new base models, presented in the previous response to Weakness 2, do not use QKNorm given their limited recalibration budget, and maintain the exact same pretraining architecture with the sole exception of skipping the application of the RoPE embeddings. Furthermore, we also note that all the recalibration hyperparameters, including model size, initialization, learning rate, and data mixture, are the same as the pretrained models. For complete transparency, we also share our entire codebase with this submission, with the full implementation, training code, and experiment configurations.
> > >
> > > Following the reviewer's feedback, we extended the first paragraph of Section 5 to provide a higher-level overview of the different design choices (as summarized in this response), and explicitly referring to our new QK norm ablation in the Appendix.

---

> > > > ### Author Response · Authors · 2025-11-26
> > > > **Authors' Response to Reviewer fTZx (Part 4/5)**
> > > >
> > > > > 2. In Table 2, what are the exact setup for SmolLM+PI/NTK/YaRN, and which token budget for the DroPE did you use?
> > > >
> > > > Following the reviewer's feedback, we realized some key evaluation details about Table 2 were previously missing, and now added them to Section 5 of the latest revision:
> > > > - For YaRN and RoPE-NTK we use the "dynamic scaling version," meaning we compute the new frequencies adaptively during generation. For PI, we instead computed the frequency scaling factor $s$ by taking the maximal evaluation length for each task.
> > > > -  For DroPE, we found that our model performed very similarly on all our recalibration budgets (given they all still comprised a very small fraction of the pretraining corpus, using "already-seen" tokens) and thus used the model trained with the smallest 30B tokens recalibration budget.
> > > >
> > > > > 3. In Appendix C.1, you mentioned a "simple QKNorm". Does that mean you modified the architecture slightly after dropping the positional encoding? What happened without? Are we able to see a detailed analysis on ablating all these variables?
> > > >
> > > > We thank the reviewer for raising this important point about architectural changes in Appendix C.1. We appreciate the opportunity to clarify the role of QKNorm [5] and verify that it is not a confounding factor for our method's effectiveness, but rather a stabilization trick we found helpful to train faster on longer budgets. In particular, was introduced strictly for optimization stability to enable training at higher learning rates. This follows recent practices in large-scale model training, such as OLMo2 [7] and Qwen3 [8], where normalization is used to stabilize gradients and lessen loss spikes rather than boost modeling capacity.
> > > >
> > > > Following the reviewer's appreciated suggestion about potential confounding effects, we conducted a controlled ablation study on the SmolLM-360M model using six configurations: three learning rates (3e-5, 3e-4, 1e-3), each trained with and without QKNorm.
> > > >
> > > > The results (summarized in Table A below) show two key findings:
> > > > - At lower learning rates (3e-5, 3e-4): DroPE works effectively without QKNorm. In fact, at the lowest learning rate (3e-5), the model without QKNorm achieves a slightly better final loss (2.713 vs. 3.102). Together with the 3e-4 setting (2.530 vs. 2.555), this shows that QKNorm does not consistently improve performance in low-volatility regimes and is not the source of our gains.
> > > > - At the highest learning rate (1e-3): The model without QKNorm becomes unstable (loss spikes, gradients explode), leading to poor convergence (loss 6.334). In contrast, adding QKNorm stabilizes training and allows us to leverage the higher learning rate to achieve the best overall performance (loss 2.496).
> > > >
> > > > Together with these results, we will also add the learning curves of training with and without QKnorm (**see Figure 12 of the latest revision**), highlighting the presence of loss spikes when training with higher learning rates akin to [7]. We believe these new results empirically demonstrate to the reader the role of QKNorm to act as a stabilizer that enables the use of a more aggressive, compute-efficient learning rate. Nonetheless, DroPE can still be applied without QKNorm by using a moderate learning rate (e.g., 3e-4), which was our default setting for all experiments except the longer Smollm-360M recalibration phases.
> > > >
> > > > As also mentioned in our previous response to Weakness 3, **we have included this ablation analysis and a clearer description of QKNorm’s role in Appendix D to stabilize training spikes beyond a certain token budget**, mirroring the results of [7]:
> > > >
> > > > Table A: Ablation study on SmolLM-360M recalibration.
> > > >
> > > > | Learning Rate | With QKNorm   | Without QKNorm | Status       |
> > > > | ------------- | -------------- | --------------- | ----------------------- |
> > > > | 1e-3 (High)   | 2.496 (Best)   | 6.334           | Unstable without Norm   |
> > > > | 3e-4 (Mid)    | 2.555          | 2.530           | Stable / Comparable   |
> > > > | 3e-5 (Low)    | 3.102          | 2.713           | Stable / Comparable  |

---

> > > > > ### Author Response · Authors · 2025-11-26
> > > > > **Authors' Response to Reviewer fTZx (Part 5/5)**
> > > > >
> > > > > **References**
> > > > >
> > > > > [1] https://huggingface.co/blog/smollm.
> > > > >
> > > > > [2] Muennighoff, Niklas, et al. "Scaling data-constrained language models." Advances in Neural Information Processing Systems 36 (2023): 50358-50376.
> > > > >
> > > > > [3] Hoffmann, Jordan, et al. "Training compute-optimal large language models." arXiv preprint arXiv:2203.15556 (2022).
> > > > >
> > > > > [4] https://huggingface.co/datasets/HuggingFaceFW/fineweb-edu
> > > > >
> > > > > [5] Henry, Alex, et al. "Query-key normalization for transformers." Findings of the Association for Computational Linguistics: EMNLP 2020. 2020.
> > > > >
> > > > > [6] Allal, Loubna Ben, et al. "SmolLM2: When Smol Goes Big--Data-Centric Training of a Small Language Model." arXiv preprint arXiv:2502.02737 (2025).
> > > > >
> > > > > [7] OLMo, Team, et al. "2 OLMo 2 Furious." arXiv preprint arXiv:2501.00656 (2024).
> > > > >
> > > > > [8] Yang, An, et al. "Qwen3 technical report." arXiv preprint arXiv:2505.09388 (2025).
> > > > >
> > > > > [9] Kočiský, Tomáš, et al. "The narrativeqa reading comprehension challenge." Transactions of the Association for Computational Linguistics 6 (2018): 317-328.
> > > > >
> > > > > [10] Hsieh, Cheng-Ping, et al. "RULER: What's the Real Context Size of Your Long-Context Language Models?." arXiv preprint arXiv:2404.06654 (2024).
> > > > >
> > > > > [11] Grattafiori, Aaron, et al. "The llama 3 herd of models." arXiv preprint arXiv:2407.21783 (2024).

---

> ### Comment · Reviewer_fTZx · 2025-11-28
>
> Thanks for the detailed response! Really great to see more detailed experiments.
>
> > According the the SmolLM blogpost [1], the additional tokens we train on (taken from FineWeb-Edu [4]) are part the SmolLM's pretraining dataset. Previous research shows that retraining on the same token has a negligible effect [2].
>
> I think it is still important to quantify this effect and understand it better. But I see that in the academia setting, one could instead argue that training the baseline for additional 20B token is too resource-intense and does not provide the best ROI.
>
> > At the highest learning rate (1e-3): The model without QKNorm becomes unstable (loss spikes, gradients explode), leading to poor convergence (loss 6.334). In contrast, adding QKNorm stabilizes training and allows us to leverage the higher learning rate to achieve the best overall performance (loss 2.496).
>
> Again, this is perhaps another resource problem. Ideally, we should properly ablate with the presence of both of them.
>
> If I were to write the paper, I would list them all out in a limitation or future work section. Personally, I think admitting to the resource constraint and showing a clear further checklist for peer researchers can earn respect and has a lot of values.
>
> Overall, I find this paper rather intriguing, and sufficiently different (in a good and justified way) from others who overly focused on tweaking architectures. I think there is likely a bit of resource constraint and time crunch for the authors, but this opens up a lot of new ideas. For example, one can think about how to smoothly bridge the RoPE phase and the NoPE phase without the drastic spike by incorporating some very mild architecture modifications. Also, this supposedly "benign" loss spike is an interesting subject in pre-training research itself. In addition, many arguments in this paper should ideally be studied in a scaling law setting, though I understand it may be difficult to secure enough compute. Nevertheless, as a fellow researcher who sees similar things in this direction, I would encourage the authors to continue exploring this direction, do more experiments and stick to the power of simplicity.
>
> I am ready to **raise the score from 6 to 7** to highlight the potential insights it can bring into this area.

---

### Official Review · Reviewer_PvVC · 2025-10-30

**Soundness:** 2
**Presentation:** 2
**Contribution:** 2
**Rating:** 2
**Confidence:** 4

**Summary:**

This paper introduces DroPE, a simple and effective method for extending the context length of pretrained language models without expensive long-context finetuning. The core idea is to remove the positional embeddings (PEs) from a fully trained model and then perform a short "recalibration" training phase. The authors argue that PEs are a crucial scaffold for efficient pretraining but inherently limit generalization to longer sequences. By dropping them post-training, DroPE achieves significant zero-shot context extension. Empirical results on a 0.5B parameter model and the SmolLM model show that DroPE outperforms established RoPE scaling methods on long-context tasks while preserving performance within the original context window.

**Strengths:**

1. The central idea of dropping positional embeddings to achieve context length extrapolation is interesting. It reframes PEs as a temporary training aid rather than a permanent architectural component.

2. The proposed DroPE method is simple to implement and demonstrates strong empirical performance, simply but effectively alleviating the OOD issue of RoPE.

**Weaknesses:**

1. The paper's central claims are not validated at a scale that reflects the current state of long-context LLMs. The experiments are limited to relatively small models (under 0.5B parameters) and a modest 2x context extension (e.g., 2048 to 4096 tokens). This is a significant limitation, as the most pressing need for context extension exists in much larger models (7B+) and for vastly longer sequences (e.g., 32k, 128k, and beyond). It is unclear if the method's effectiveness and training stability would hold when extending context by 10x or 100x, where challenges like attention dilution and loss of positional signal become far more severe. The strong claims of the paper require evidence at a more demanding scale to be fully convincing.

2. While the paper provides extensive analysis, a notable portion of it lacks novelty and feels disconnected from the proposed method. The analysis in Section 4, which details the failure of RoPE-scaling methods due to the compression of low frequencies, largely reiterates well-established findings from the original papers on YaRN, NTK-RoPE, and LongRoPE. Furthermore, these theoretical insights do not directly inform the specific design of the DroPE recalibration process, such as the required duration or optimal hyperparameters. The theory explains why a problem exists but offers little guidance on how to best implement the proposed solution.

3. The theoretical justifications in Section 3, intended to prove the necessity of PEs during training, rely on arguments that feel trivial. For example, Proposition 3.2 proves that a NoPE transformer's gradients vanish on an artificial sequence of identical tokens. This is an unrealistic edge case that has little bearing on training with diverse, real-world data. Spending significant space on such formalisms seems to over-justify a widely accepted premise (that PEs help training) and detracts from what could have been a more focused empirical investigation into the DroPE method itself.

**Questions:**

1. How do you expect DroPE to perform when scaling to much larger models (e.g., 7B+) and extending the context by a much larger factor (e.g., from 4k to 128k)? Do you anticipate any new optimization challenges?

2. Have you evaluated DroPE on tasks that are highly sensitive to precise token positions (e.g., Passage reranking in HELMET)? Is there a trade-off where long-context generalization comes at the cost of fine-grained positional awareness?

[1] HELMET: How to Evaluate Long-context Language Models Effectively and Thoroughly

3. What is the intuition for what the model learns during recalibration? Is it primarily learning to infer relative positions from the causal mask, and have you observed corresponding changes in attention patterns?

---

> ### Author Response · Authors · 2025-11-25
> **Authors' Response to Reviewer PvVC (Part 1/5)**
>
> We thank the reviewer for their thoughtful comments and for finding our idea "interesting", praising the simplicity and effectiveness of our method. We also appreciate the constructive feedback and concerns, and address them here:
>
> **Weaknesses**
> > 1. The paper's central claims are not validated at a scale that reflects the current state of long-context LLMs. The experiments are limited to relatively small models (under 0.5B parameters) and a modest 2x context extension (e.g., 2048 to 4096 tokens). This is a significant limitation, as the most pressing need for context extension exists in much larger models (7B+) and for vastly longer sequences (e.g., 32k, 128k, and beyond). It is unclear if the method's effectiveness and training stability would hold when extending context by 10x or 100x, where challenges like attention dilution and loss of positional signal become far more severe. The strong claims of the paper require evidence at a more demanding scale to be fully convincing.
>
> To address this, we have now **added new experiments and an extended analysis** to Section 5 of the updated manuscript, and provide a larger **variety of base models** at even **larger scales**, more **comprehensive evaluation** with **longer extrapolation distances**, and **additional baselines**. We divide our response into two parts, which elaborate on each of these components and summarize the main analysis and results:
>
> **Part 1: Base models tested.** In Section 5 of our initial submission, we run experiments considering two main settings using different model scales:
> - We used a 500M parameter RoPE transformer trained to $2\times$ the chinchilla optimal rate to show that DroPE can be integrated as a *zero cost* extension at the end of standard pretraining pipelines, in order to perform targeted controlled experiments and compare DroPE with prior baselines.
> - We used the SmolLM-360M model pretrained on 600B tokens to show that DroPE can be applied to *adapt LLMs in the wild* at a fraction of their original training budget and study how far this can be pushed in terms of sample efficiency.
>
> Following the reviewer's feedback, **we have added new experiments to the latest revision, considering two additional pretrained LLMs with increasing numbers of parameters and pretraining context lengths**:
>
> First, we considered a larger 1.7B SmolLM model that was trained on 1T tokens. We applied DroPE and then recalibrated this model on only 20B tokens, 2% of the original training budget. As shown in the table below, despite the much lower recalibration budget in relative terms, our findings for this larger model are consistent with the ones obtained using the 360M variant, with DroPE outperforming other context extension techniques.
>
>   | Dataset       | Base  | NTK   | YaRN  |   PI  |   DroPE   |
>   |---------------|-------|-------|-------|-------|-----------|
>   | MultifieldQA  | 4.12  | 27.58 | 27.60 | 4.12  | **32.18** |
>   | MusiQue       | 0.50  | 3.37  | 3.90  | 0.50  | **7.53**  |
>   | GovReport     | 4.70  | 24.65 | 17.19 | 4.70  | **24.77** |
>
> Second, we considered Llama2-7B [2], originally pretrained on 4T tokens with 4096 context length, recalibrated once again for 20B tokens, this time only 0.5% of the original budget. We chose this model as the Llama family was suggested in the reviewer's question 3 below. We note that due to model scale and the increased context, merely adapting this model for 20B tokens already took more than 2000 H100 GPU hours. Furthermore, as the team behind Llama2-7B did not disclose the details of their pretraining data, we still resorted to the SmolLM training corpus. Nonetheless, despite having had to push the recalibration phase budget to the minimum in relative terms and not having access to the Llama closed-source corpus, we found our Llama7B-DroPE model still outperforms all RoPE-context extension scaling baselines on LongBench.
>
>   | Dataset      | Base  | NTK   | YaRN  | DroPE |
>   |-------------|-------|-------|-------|-------|
>   | MultifieldQA| 17.26 | 21.81 | 23.13 | **25.90** |
>   | MusiQue     | 10.43 | 10.91 | 7.65  | **12.88** |
>   | GovReport   | 32.41 | 32.91 | 26.65 | **39.47** |
>
> We hope these new results will significantly reinforce the evidence of the generality and scalability of DroPE, and that our efforts will help address the reviewer's feedback about the diversity and scale of base models.

---

> > ### Author Response · Authors · 2025-11-25
> > **Authors' Response to Reviewer PvVC (Part 2/5)**
> >
> > **Part 2: Extrapolation distances considered and comprehensiveness of benchmarks.** Our original evaluation considered seven pretraining tasks [1] for in-context evaluation, three synthetic tasks from the RULER benchmark [3] where we chose to evaluate at 2x our model's training context length, and five challenging LongBench tasks with up to 32K tokens corresponding to 16x of our model's original context length. As suggested in the reviewer's feedback, **we extended our evaluation settings** with a detailed analysis of how varying the context extension lengths affects performance, and **evaluated our method on additional tasks and benchmarks**.
> >
> > First, to analyze how varying context length affects performance at a granular level beyond 2x, we started by evaluating our models on NIAH tasks of up to 8x of our original context length:
> >
> >   | Context length            |   DroPE   |  NTK  | YaRN  |
> >   |---------------------------|-----------|-------|-------|
> >   | 4K (2x original ctx)      | **74.92** | 29.84 | 48.25 |
> >   | 8K (4x original ctx)      | **55.00** | 14.37 | 25.62 |
> >   | 16K (8x original ctx)     | **52.50** |  7.19 | 12.18 |
> >
> > Additionally, we also recorded granular performances by considering the scores attained by the different context extension strategies, by partitioning LongBench problems by their input context lengths. To remove confounding factors, we focused this analysis on two tasks that include problems across all our considered context length evaluation brackets up to x8 and x16:
> >
> > **Multifield QA**
> > | model        | 0-4K (0-2x ctx) | 4-8K (2-4x ctx) | 8-16K (4-8x ctx) |
> > | ------------ | --------------- | --------------- | ---------------- |
> > | SmolLM-DroPE | 32.82           | **24.73**       | **30.07**        |
> > | SmolLM-NTK   | **34.25**       | 22.3            | 21.63            |
> > | SmolLM-YaRN  | 33.96           | 22.91           | 20.08            |
> >
> > **MusiQue**
> >
> > | model        | 0-4K (0-2x ctx) | 4-8K (2-4x ctx) | 8-16K (4-8x ctx) | 16-32K (8-16x ctx) |
> > | ------------ | --------------- | --------------- | ---------------- | ------------------ |
> > | SmolLM-DroPE | **50**          | 6.11            | **8.05**         | **16.67**          |
> > | SmolLM-NTK   | 0               | 4.36            | 3.36             | 0.0                |
> > | SmolLM-YaRN  | 0.0             | **19.68**       | 3.13             | 7.14               |
> >
> > As shown by our new analysis results above, DroPE significantly outperforms other RoPE-extension baselines across all extension factors. Interestingly, these results also reveal that beyond a 4x context extension factor, DroPE's performance appears to stabilize while the other methods sharply deteriorate. This is most evident on the synthetic NIAH tasks, where we are able to precisely control the distance to the information to retrieve for success. In this instance, we see DroPE's performance drops by only 2.5% (from 55% to 52.5%) when doubling the context length from 4x to 8x. In contrast, when applying NTK and YaRN, model performance halves (14.37% to 7.19% for NTK, and 25.62% to 12.18% for YaRN). We believe these results are in line with our motivating observations and theory: zero-shot context extension inherently conflicts with the presence of RoPE positional encodings (see Section 4), with DroPE being a practical method to break this paradigm without losing RoPE's substantial training benefits.
> >
> > Finally, we note that, to our knowledge, no other method reported 10-100x extension *when evaluated zero-shot* on meaningful long context tasks that require recall. To our knowledge, all prior methods that could perform non-trivial tasks on 128K sequences have some form of long context training, with a cost orders of magnitude superior to DroPE. Our method aims to break the bottleneck of long-context training, which is highly computationally expensive.

---

> > > ### Author Response · Authors · 2025-11-25
> > > **Authors' Response to Reviewer PvVC (Part 3/5)**
> > >
> > > > 2. While the paper provides extensive analysis, a notable portion of it lacks novelty and feels disconnected from the proposed method. The analysis in Section 4, which details the failure of RoPE-scaling methods due to the compression of low frequencies, largely reiterates well-established findings from the original papers on YaRN, NTK-RoPE, and LongRoPE. Furthermore, these theoretical insights do not directly inform the specific design of the DroPE recalibration process, such as the required duration or optimal hyperparameters. The theory explains why a problem exists but offers little guidance on how to best implement the proposed solution.
> > >
> > > We thank the reviewer for the thoughtful comment. Following their feedback, we added an extended discussion to better highlight the novel aspects of our analysis by explicitly contrasting our theoretical results to previous analyses and better connecting them to our design choices in Section 4. Moreover, we also added to the latest revision **new ablations and results regarding the duration of the DroPE recalibration phase** in Section 5 and the Appendix.
> > >
> > > **On the novelty and role of the analysis**
> > > While prior RoPE-extension works (e.g., YaRN, NTK-RoPE, LongRoPE) discuss RoPE frequencies, our analysis in Section 4 is not a restatement of their results but a unified critique of the RoPE-scaling paradigm itself:
> > > - We explicitly analyze existing RoPE-scaling methods and show that their compression of low frequencies induces **a shift of attention mass** at long ranges, which we empirically link to poor zero-shot long-context retrieval (e.g., needle-in-a-haystack), even when perplexity remains stable. This connection between low-frequency compression and retrieval failures is, to our knowledge, not provided in prior work.
> > > - We then prove that such low-frequency compression is inevitable for any RoPE extension that rescales frequencies to keep positional phases in-distribution. Thus, Section 4 does not merely re-derive the behavior of one or two methods; it identifies a structural limitation of the entire class of RoPE-scaling approaches.
> > > - Finally, we show empirically that popular scaling schemes implicitly behave like a cropping strategy: they maintain perplexity while effectively discarding information beyond an effective window, which further explains why they underperform on retrieval-oriented benchmarks. This perspective directly highlights the failure mode we target.
> > >
> > > **On how the analysis informs DroPE and its recalibration design**
> > > Sections 3 and 4 are designed to justify DroPE’s architectural and training procedure choices, rather than to produce a formula for hyperparameters:
> > > - Section 4 provides the _negative_ design result: any RoPE-scaling approach must compress low frequencies and thus harm long-range semantic attention. This motivates our decision to abandon RoPE entirely at inference time, instead of proposing yet another RoPE scaling method.
> > > - Section 3 provides the _positive_ design result: it explains why RoPE is particularly beneficial _early_ in training and why pure NoPE training is notoriously difficult. This is a phenomenon that, to our knowledge, has not been given a theoretical explanation. This motivates our choice to pretrain with RoPE and only then remove it, rather than training a NoPE model from scratch.
> > > - Regarding the duration and hyperparameters of recalibration: as in other long-context extension methods, the exact token budget is determined empirically. What our theory contributes is: (i) recalibration should occur after RoPE pretraining, when the model has already learned strong semantic structure, and (ii) RoPE’s main advantage is in creating attention non-uniformity, suggesting that after non-uniformity is achieved during training, a relatively short recalibration phase suffices.
> > >
> > > **Empirically examining the RoPE recalibration process**
> > > In order to empirically confirm the theoretical implications of our work, we have added new experiments varying at which stage during training to activate recalibration, with four total recipes:
> > > - Dropping PEs at step 0 (NoPE transformer),
> > > - Dropping PEs at step 8K,
> > > - Dropping PEs at step 14K,
> > > - Dropping PEs at step 16K (RoPE transformer).
> > >
> > > We report here the final validation perplexity after training:
> > > |DroPE @ 0K|DroPE @ 8K|DroPE @ 14K|DroPE @ 16K|
> > > |-|-|-|-|
> > > |23.77|22.42|21.73|21.72|
> > >
> > > We find the results of this ablation further strengthen our theoretical observation that DroPE should be integrated **further into training**. This is supported by our analysis in Section 3, which suggests that NoPE transformers struggle to train efficiently and that dropping the positional encoding works best at the very end to keep harnessing the "faster" training of RePE.
> > >
> > > We hope this new discussion and experiments will help clarify the contribution of our analysis and provide a more explicit, direct link between our theory and our design choices.

---

> > > > ### Author Response · Authors · 2025-11-26
> > > > **Authors' Response to Reviewer PvVC (Part 4/5)**
> > > >
> > > > > 3. The theoretical justifications in Section 3, intended to prove the necessity of PEs during training, rely on arguments that feel trivial. For example, Proposition 3.2 proves that a NoPE transformer's gradients vanish on an artificial sequence of identical tokens. This is an unrealistic edge case that has little bearing on training with diverse, real-world data. Spending significant space on such formalisms seems to over-justify a widely accepted premise (that PEs help training) and detracts from what could have been a more focused empirical investigation into the DroPE method itself.
> > > >
> > > > We agree with the reviewer that Proposition 3.2 considers an edge case (a sequence of identical tokens). This result is intentionally presented as a warm-up example that isolates the core mechanism: in a NoPE transformer, uniform inputs lead to uniform attention, resulting in vanishing gradients for the positional structure and, therefore, no learning signal for positional bias. We do not rely on this edge case for our main conclusions. We note that many mechanistic analysis papers use similar constructions to isolate specific behaviors of the attention mechanism. For instance, [1] analyzes RoPE using settings with constant vectors to understand its decay and frequency behavior at a mechanical level. Similarly, [2] relies on controlled synthetic probes to study how NoPE models acquire positional awareness. Our Proposition 3.2 plays a similar role as a transparent construction that makes the failure mode of NoPE explicit, and then Theorem 3.4 shows that this limitation persists.
> > > >
> > > > However, we would like to clarify that the substantive theoretical contribution of Section 3 is Theorem 3.4, which _does not_ assume identical tokens and instead works under realistic initialization conditions. It shows that, for NoPE transformers initialized with small-variance i.i.d. embeddings (the common practice in large-scale LM training), the hidden states and attention maps remain close to uniform across arbitrary sequences, and the induced positional bias and its gradients remain uniformly small, independent of sequence length. This formalizes an empirical phenomenon: even though NoPE transformers can express positional information, they start with nearly uniform attention in which positional structure is hard to learn, whereas RoPE transformers immediately inject strong positional signals. **In the latest revision, we modified Section 3 to make this hierarchy more explicit by clearly presenting Proposition 3.2 as an illustrative toy example and emphasizing Theorem 3.4 as the main theoretical result.**
> > > >
> > > > Finally, we note that the purpose of Section 3 is to mechanistically explain why RoPE makes training easier compared to NoPE, thereby justifying our design choice. Most of the analysis of NoPE transformers in previous papers [2, 3] is focused on _expressivity_ and not _optimization_. We are not aware of any explanation that specifically targets NoPE's _training_ difficulties. **Following the reviewer's comments, we extended Section 6 in order to make this critical aspect of our NoPE analysis more explicit.**

---

> > > > > ### Author Response · Authors · 2025-11-26
> > > > > **Authors' Response to Reviewer PvVC (Part 5/5)**
> > > > >
> > > > > **Questions**
> > > > > > 1. How do you expect DroPE to perform when scaling to much larger models (e.g., 7B+) and extending the context by a much larger factor (e.g., from 4k to 128k)? Do you anticipate any new optimization challenges?
> > > > >
> > > > > See our answer to weakness 1.
> > > > >
> > > > > > 2. Have you evaluated DroPE on tasks that are highly sensitive to precise token positions (e.g., Passage reranking in HELMET)? Is there a trade-off where long-context generalization comes at the cost of fine-grained positional awareness?
> > > > >
> > > > > We thank the reviewer for this interesting question. Since our method is applied immediately after pretraining, our resulting model is not instruction-tuned and thus can struggle at a complex task like passage re-ranking. Still, when we evaluated Llama2-DroPE on passage re-ranking at 2x context size, it outperforms the base model. Results are reported as pass @ 1.
> > > > >
> > > > > |Llama2-7B| Llama2-7B-DroPE|
> > > > > |----|--------|
> > > > > |0.0|**8.13**|
> > > > >
> > > > > We believe that understanding the performance of NoPE architectures on tasks that are sensitive to token position is an interesting future direction, enabled by our introduction of performant NoPE LMs through DroPE.
> > > > >
> > > > > > 3. What is the intuition for what the model learns during recalibration? Is it primarily learning to infer relative positions from the causal mask, and have you observed corresponding changes in attention patterns?
> > > > >
> > > > > Following the reviewer's question, we have added new experiments and analysis to the latest revision, investigating the attention maps of our own pretrained 500B RoPE- and DroPE-transformers LMs. We start by measuring the correlation between the attention maps of both models over 100 randomly sampled sequences of length 32. We find that there is a positive correlation of 0.68. A more fine-grained qualitative analysis shows that the match is weaker for positional heads, i.e., ones that attend consistently based on relative position, as ordering information has to be inferred from "bos" tokens and capitalized letters. On the other hand, for semantic heads, the correlation is higher, showing how the semantic notions are largely preserved. Furthermore, to provide readers with visual intuition for this analysis, **we will add direct visualizations of the attention maps, precisely showing this phenomenon**, highlighting the recalibration focus on modifying positional heads that rely the most on positional embeddings.
> > > > >
> > > > > ---
> > > > >
> > > > > **References**
> > > > >
> > > > > [1] Barbero, Federico, et al. "Round and Round We Go! What makes Rotary Positional Encodings useful?" The Thirteenth International Conference on Learning Representations.
> > > > >
> > > > > [2] Haviv, Adi, et al. "Transformer language models without positional encodings still learn positional information." arXiv preprint arXiv:2203.16634 (2022).
> > > > >
> > > > > [3] Kazemnejad, Amirhossein, et al. "The impact of positional encoding on length generalization in transformers." Advances in Neural Information Processing Systems 36 (2023): 24892-24928.

---

### Official Review · Reviewer_iW8i · 2025-10-31

**Soundness:** 2
**Presentation:** 3
**Contribution:** 2
**Rating:** 4
**Confidence:** 4

**Summary:**

This paper proposes a novel method to extend the context length of pretrained language models by removing RoPE after pretraining and conducting a short recalibration phase. The authors argue that positional embeddings are crucial for training convergence but harms zero-shot generalization to longer contexts.

**Strengths:**

1. Novel and counterintuitive hypothesis where positional embeddings might not be necesssary throughout a model's lifecycle.
2. Treating PEs (Positional Embeddings) as "training scaffolds" that can be removed is elegant and very good for downstream use cases such as long context fine-tuning and kv-cached inference performance.
3. Good theoretical contributions on why PEs are necessary during training and why they can be removed after.
4. Strong potential for practical impact because of how simple this method is. Furthermore, reduces the inductive biases of LLMs after removing RoPE.

**Weaknesses:**

1. Experiments on small models might not scale well to larger models. Could be possible to take existing large LMs and recalibrate them with DroPE.
2. No ablations or thorough experimentation on the recalibration phase. Naively removing PEs might work but given the theory there should be better ways. Furthermore, recalibration cost is not explored either.
3. Dubious claim that YaRN cannot extrapolate to longer contexes, directly contradicting the original paper and results from the industry (DeepSeek R1, Qwen3, GPT-OSS).
4. Lack of comparisons against more recent length generalization work (LongRope 2, sparse attention, etc.)
5. Only test at 2k context lengths, most methods now test at 128k+.
6. Specific solution to autoregressive LLMs, does not generalize to diffusion models.

**Questions:**

1. Why was YaRN not able to extrapolate to 2x context length? This might be a mistake since Qwen3 uses YaRN without finetuning and has a perfect score in the needle-in-a-haystack benchmark.
2. Do you know if DroPE is better than YaRN because it has more training time during the recalibration phase? Can you compare it against YaRN but with the same recalibration phase?
3. Do you think that gradually removing the RoPE during the recalibration phase be better than outright removing it in a single step? It's highly likely that DroPE is equivalent to setting a RoPE Linear scaling to infinity.
4. What happens with DroPE's performance at higher context scaling lengths? 4x, 8x, etc?

---

> ### Author Response · Authors · 2025-11-25
> **Authors' Response to Reviewer iW8i (Part 1/5)**
>
> We thank the reviewer for their thoughtful comments and for praising our paper’s novel and counterintuitive results, elegant and useful approach, theoretical contributions, and strong potential for practical impact. We also appreciate the constructive feedback and concerns, and address them here:
>
> **Weaknesses**
> > 1. Experiments on small models might not scale well to larger models. Could be possible to take existing large LMs and recalibrate them with DroPE.
>
> In Section 5 of our initial submission, we run experiments considering two main settings using different model scales:
> - We used a 500M parameter RoPE transformer trained to $2\times$ the chinchilla optimal rate to show that DroPE can be integrated as a *zero cost* extension at the end of standard pretraining pipelines. The goal of this setup is to perform targeted controlled experiments and compare DroPE with prior baselines.
> - We used the SmolLM-360M model pretrained on 600B tokens to show that DroPE can be applied to *adapt LLMs in the wild* at a fraction of their original training budget and study how far this can be pushed in terms of sample efficiency.
>
> Following the reviewer's feedback, we have **added new experiments** to the latest revision, considering **two additional pretrained LLMs** with increasing **numbers of parameters** and pretraining **context lengths**:
>
> First, we considered a larger 1.7B SmolLM model that was trained on 1T tokens. We applied DroPE and then recalibrated this model on only 20B tokens, 2% of the original training budget. As shown in the table below, despite the much lower recalibration budget in relative terms, our findings for this larger model are consistent with the ones obtained using the 360M variant, with DroPE outperforming other context extension techniques.
>
>   | Dataset       | Base  | NTK   | YaRN  |   PI  |   DroPE   |
>   |---------------|-------|-------|-------|-------|-----------|
>   | MultifieldQA  | 4.12  | 27.58 | 27.60 | 4.12  | **32.18** |
>   | MusiQue       | 0.50  | 3.37  | 3.90  | 0.50  | **7.53**  |
>   | GovReport     | 4.70  | 24.65 | 17.19 | 4.70  | **24.77** |
>
> Second, we considered Llama2-7B, originally pretrained on 4T tokens with 4096 context length, recalibrated once again for 20B tokens, which this time comprise only 0.5% of the original budget. We note that due to model scale and the increased context, merely adapting this model for 20B tokens already took more than 2000 H100 GPU hours. Furthermore, as the team behind Llama2-7B did not disclose the details of their pretraining data, we still resorted to the SmolLM training corpus. Nonetheless, despite having had to push the recalibration phase budget to the minimum in relative terms and not having access to the Llama closed-source corpus, we found our Llama7B-DroPE model still outperforms all RoPE-context extension scaling baselines on LongBench.
>
>   | Dataset      | Base  | NTK   | YaRN  | DroPE |
>   |-------------|-------|-------|-------|-------|
>   | MultifieldQA| 17.26 | 21.81 | 23.13 | **25.90** |
>   | MusiQue     | 10.43 | 10.91 | 7.65  | **12.88** |
>   | GovReport   | 32.41 | 32.91 | 26.65 | **39.47** |
>
> We hope these new results will significantly reinforce the evidence of the generality and scalability of DroPE, and that our efforts will help address the reviewer's feedback about the diversity and scale of base models.

---

> > ### Author Response · Authors · 2025-11-25
> > **Authors' Response to Reviewer iW8i (Part 2/5)**
> >
> > > 2. No ablations or thorough experimentation on the recalibration phase. Naively removing PEs might work but given the theory there should be better ways. Furthermore, recalibration cost is not explored either.
> >
> > We thank the reviewer for pointing out the importance of exploring the cost of the recalibration phase and ablating its components. In the revised manuscript, we highlight the **different recalibration recipes we considered** for applying DroPE to LMs in the wild, as well as a **new ablation on the precise time where we apply DroPE** in the training from scratch setup in Section 5.1.
> >
> > **Recalibrating LMs in the wild**
> > We agree with the reviewer that ablations are important in order to find the optimal recalibration scheme, and point the reviewer to Figure 10 in our paper, which studies the performance of three different recalibration recipes on 30B, 60B, and 120B tokens, reporting average in-context performance throughout training every 5B steps:
> >
> > - 30B recipe
> >   |                | 0B    | 5B    | 10B   | 15B   | 20B   | 25B   | 30B   |
> >   |----------------|-------|-------|-------|-------|-------|-------|-------|
> >   | Avg. eval score | 35.93 | 50.23 | 51.12 | 51.20 | 51.62 | 52.22 | 52.55 |
> > - 60B recipe
> >   |                | 0B    | 5B    | 10B   | 15B   | 20B   | 25B   | 30B   | 35B   | 40B   | 45B   | 50B   | 55B   | 60B   |
> >   |----------------|-------|-------|-------|-------|-------|-------|-------|-------|-------|-------|-------|-------|-------|
> >   | Avg. eval score | 35.93 | 49.97 | 50.57 | 49.78 | 50.85 | 50.60 | 50.45 | 50.83 | 50.97 | 51.95 | 51.92 | 52.72 | 52.98 |
> > - 120B recipe
> >   |                | 0B    | 5B    | 10B   | 15B   | 20B   | 25B   | 30B   | 35B   | 40B   | 45B   | 50B   | 55B   | 60B   | 65B   | 70B   | 75B   | 80B   | 85B   | 90B   | 95B   | 100B  | 105B  | 110B  | 115B  | 120B  |
> >   |----------------|-------|-------|-------|-------|-------|-------|-------|-------|-------|-------|-------|-------|-------|-------|-------|-------|-------|-------|-------|-------|-------|-------|-------|-------|-------|
> >   | Avg. eval score | 35.93 | 50.23 | 50.50 | 50.63 | 51.13 | 50.93 | 51.75 | 51.62 | 51.68 | 51.70 | 51.78 | 52.30 | 52.13 | 52.18 | 52.47 | 52.27 | 53.00 | 53.05 | 53.15 | 53.02 | 53.15 | 53.25 | 53.15 | 53.48 | 53.43 |
> >
> > As summarized in the tables above, we find that we can restore **91.9%** of the in-context performance after only 5B steps, and that a 60B recipe can match in-context performance exactly. In terms of cost, a recalibration step is identical to a pretraining step since it is performed at the same context length. Therefore, for SmolLM-360M, 5B of the tokens is equivalent to a negligible **0.83%** of the pretraining cost.
> >
> > **Recalibration as part of pretraining**
> > In this setup, we train a 500M parameter transformer on 16B tokens and remove its PEs **during training**. Following the reviewer's feedback, we have added new experiments varying at which stage during training to activate recalibration, with four total recipes:
> > - Dropping PEs at step 0 (NoPE transformer),
> > - Dropping PEs at step 8K,
> > - Dropping PEs at step 14K,
> > - Dropping PEs at step 16K (RoPE transformer).
> >
> > We report here the final validation perplexity after training:
> > |DroPE @ 0K|DroPE @ 8K|DroPE @ 14K|DroPE @ 16K|
> > |-|-|-|-|
> > |23.77|22.42|21.73|21.72|
> >
> > We find the results of this ablation further strengthen our theoretical observation that DroPE should be integrated **further into training**. This is supported by our analysis in Section 3, which suggests that NoPE transformers struggle to train efficiently and that dropping the positional encoding works best at the very end to keep harnessing the "faster" training of RoPE transformers. We further stress that in this setup, **there is no additional cost to DroPE**, as the total number of training steps stays constant and removing the positional embedding even makes training slightly faster (due to removing the rotation operations during attention).

---

> > > ### Author Response · Authors · 2025-11-25
> > > **Authors' Response to Reviewer iW8i (Part 3/5)**
> > >
> > > > 3. Dubious claim that YaRN cannot extrapolate to longer contexts, directly contradicting the original paper and results from the industry (DeepSeek R1, Qwen3, GPT-OSS).
> > >
> > > We thank the reviewer for pointing out this counterintuitive statement. This point is at the heart of our paper, and we are happy to clarify it here. While YaRN and other similar context extension methods are used by some open source LLMs, it is added as part of the _mid-training_ stage, which includes expensive **long-context finetuning** on progressively longer sequences. In contrast, our method performs recalibration at the **original context length** and can be **integrated at no extra cost during pretraining**. We emphasize that this is a _crucial_ difference, since both the runtime and the memory cost of training grow _quadratically_ with sequence length.
> > >
> > > More generally, it has been shown that YaRN can preserve perplexity at longer context sizes, but when it’s evaluated *zero-shot*, i.e., without a long-context mid-training phase, it often fails on retrieval tasks. Figure 5 in our paper demonstrates this exact phenomenon, showing that for needle-in-a-haystack tasks, zero-shot YaRN behaves similarly to context cropping. Similar results were also reported in the long-context modelling literature, e.g., in [3]. Additionally, inspired by the reviewer's comment, to further demonstrate this phenomenon, **we have added needle-in-a-haystack experiments for the strong Qwen3 model**, extended zero-shot with YaRN** (See our response to question 1 for the details).
> > >
> > > > 4. Lack of comparisons against more recent length generalization work (LongRope 2, sparse attention, etc.)
> > >
> > > In our original paper, we compared DroPE with zero-shot PI, YaRN, and NTK-aware context extension, together with NoPE and RNoPE-SWA trained in our same 20B controlled model budgets. Following the reviewer's feedback about additional baselines, **we have now added two additional baselines**, including the suggested **LongRoPE2** [8] context extension** (which we also incorporated zero-shot) and **AliBi** [9], which we trained from scratch using the same protocol as for our controlled experiments:
> > >
> > > **SmolLM post-training adaptation**
> > > | Context length            |   DroPE   |  NTK  | YaRN  | LongRoPE2 |
> > > |---------------------------|-----------|-------|-------|-----------|
> > > | 4K (2x original ctx)      | **74.92** | 29.84 | 48.25 |   44.20   |
> > > | 8K (4x original ctx)      | **55.00** | 14.37 | 25.62 |   26.20   |
> > > | 16K (8x original ctx)     | **52.50** |  7.19 | 12.18 |   16.45   |
> > >
> > > **Models trained from scratch**
> > > | Method                                  | Multi-Query | Multi-Key | Multi-Value |
> > > |-----------------------------------------|------------:|----------:|------------:|
> > > | RoPE transformer                        |        0.0  |      0.0  |        0.0  |
> > > | RoPE transformer + PI                   |        0.0  |      0.0  |        0.0  |
> > > | RoPE transformer + RoPE-NTK             |       21.1  |     19.4  |       16.5  |
> > > | RoPE transformer + YaRN                 |       17.8  |      0.5  |       14.6  |
> > > | RNoPE-SWA transformer                   |        5.2  |     25.6  |       20.6  |
> > > | NoPE transformer                        |        9.2  |     36.2  |       21.4  |
> > > | ALiBi transformer                       |        5.2  |      0.0  |        1.1  |
> > > | DroPE transformer                       |     **28.0**  |     **41.6**  |   **23.3**  |
> > >
> > > Our results show that, when applied zero-shot, LongRope2 behaves similarly to YaRN, surpassing ROPE-NTK context extension but still lagging significantly behind DroPE. We find that these results are consistent with the prior literature analyzing the insufficiencies of RoPE-based context extension methods [3], without the very expensive mid-training stage that our new method seeks to overcome (LongRope2 was originally shown to be effective only with a long mid-training phase using up to 200K context length). We note that ALiBi models have been known to struggle with needle-in-a-haystack tasks, and these results are consistent with past literature [10]. We posit that this is due to the persistent positional bias that, together with the attention softmax, would inevitably downweigh the semantic attention heads required for NIAH tasks.

---

> > > > ### Author Response · Authors · 2025-11-25
> > > > **Authors' Response to Reviewer iW8i (Part 4/5)**
> > > >
> > > > > 5. Only test at 2k context lengths, most methods now test at 128k+.
> > > >
> > > > Our original evaluation considered seven pretraining tasks [11] for in-context evaluation, three synthetic tasks from the RULER benchmark [12] where we chose to evaluate at 2x our model's training context length, and five challenging LongBench tasks with up to 32K tokens corresponding to 16x of our model's original context length. We note that even strong models, trained on trillions of tokens, e.g., Llama3 [7], only evaluate on 2 tasks from this suite. As seen by the following breakdown, DroPE excels on the longer task instances compared to the RoPE-extension baselines. As suggested in the reviewer's feedback, **we extended our evaluation settings with a detailed analysis of how varying the context extension lengths affects performance beyond 2K**.
> > > >
> > > > To analyze how varying context length affects performance at a granular level beyond 2x, we started by evaluating our models on NIAH tasks of up to 8x of our original context length:
> > > >
> > > >   | Context length            |   DroPE   |  NTK  | YaRN  |
> > > >   |---------------------------|-----------|-------|-------|
> > > >   | 4K (2x original ctx)      | **74.92** | 29.84 | 48.25 |
> > > >   | 8K (4x original ctx)      | **55.00** | 14.37 | 25.62 |
> > > >   | 16K (8x original ctx)     | **52.50** |  7.19 | 12.18 |
> > > >
> > > > Additionally, we also recorded granular performances by considering the scores attained by the different context extension strategies, by partitioning LongBench problems by their input context lengths. To remove confounding factors, we focused this analysis on two tasks that include problems across all our considered context length evaluation brackets up to x8 and x16:
> > > >
> > > > **Multifield QA**
> > > > | model        | 0-4K (0-2x ctx) | 4-8K (2-4x ctx) | 8-16K (4-8x ctx) |
> > > > | ------------ | --------------- | --------------- | ---------------- |
> > > > | SmolLM-DroPE | 32.82           | **24.73**       | **30.07**        |
> > > > | SmolLM-NTK   | **34.25**       | 22.3            | 21.63            |
> > > > | SmolLM-YaRN  | 33.96           | 22.91           | 20.08            |
> > > >
> > > > **MusiQue**
> > > >
> > > > | model        | 0-4K (0-2x ctx) | 4-8K (2-4x ctx) | 8-16K (4-8x ctx) | 16-32K (8-16x ctx) |
> > > > | ------------ | --------------- | --------------- | ---------------- | ------------------ |
> > > > | SmolLM-DroPE | **50**          | 6.11            | **8.05**         | **16.67**          |
> > > > | SmolLM-NTK   | 0               | 4.36            | 3.36             | 0.0                |
> > > > | SmolLM-YaRN  | 0.0             | **19.68**       | 3.13             | 7.14               |
> > > >
> > > > As shown by our new analysis results above, DroPE significantly outperforms other RoPE-extension baselines across all extension factors. Interestingly, these results also reveal that beyond a 4x context extension factor, DroPE's performance appears to stabilize while the other methods sharply deteriorate. This is most evident on the synthetic NIAH tasks, where we are able to precisely control the distance to the information to retrieve for success. In this instance, we see DroPE's performance drops by only 2.5% (from 55% to 52.5%) when doubling the context length from 4x to 8x. In contrast, when applying NTK and YaRN, model performance halves (14.37% to 7.19% for NTK, and 25.62% to 12.18% for YaRN). We believe these results are in line with our motivating observations and theory: zero-shot context extension inherently conflicts with the presence of RoPE positional encodings (see Section 4), with DroPE being a practical method to break this paradigm without losing RoPE's substantial training benefits.
> > > >
> > > > Finally, we note that, to our knowledge, all prior methods that could perform non-trivial tasks on 128K sequences have some form of long context training, with a cost orders of magnitude superior to DroPE. Our method aims to break the bottleneck of long-context training, which is highly computationally expensive. Therefore, we focus on the zero-shot setting in our experiments.
> > > >
> > > > > 6. Specific solution to autoregressive LLMs, does not generalize to diffusion models.
> > > >
> > > > We thank the reviewer for raising this point. We emphasize that our methods work for all transformer-based methods that use positional embeddings, particularly for diffusion LLMs such as [4, 5]. **Following the reviewer's feedback, we added this clarification and highlighted how diffusion LMs are compatible with DroPE and pose an interesting direction for future work to Section 7 of the revised manuscript**.

---

> > > > > ### Author Response · Authors · 2025-11-25
> > > > > **Authors' Response to Reviewer iW8i (Part 5/5)**
> > > > >
> > > > > **Questions**
> > > > >
> > > > > > 1. Why was YaRN not able to extrapolate to 2x context length? This might be a mistake since Qwen3 uses YaRN without finetuning and has a perfect score in the needle-in-a-haystack benchmark.
> > > > >
> > > > > As mentioned in our answer to weakness 3, the lack of zero-shot generalization of YaRN, specifically for needle-in-a-haystack tasks, has been reported in the past [3]. The Qwen3 technical report [2] mentions that, in addition to YaRN, the model does go through a long context training phase on sequences of length up to 32K with a RoPE basis change and even integrates Dual-Chunk attention at inference time. Following the reviewer's feedback, **we added new experiments** to the latest revision, where we **evaluate the base Qwen3-1.7B model, extended with YaRN alone**, on the single query needle-in-a-haystack tasks from the RULER benchmark. We run on 32K (1x), 64K (2x), and 128K (4x) sequence lengths with a YaRN scaling factor of 4:
> > > > >
> > > > > |32K|64K|128K|
> > > > > |-|-|-|
> > > > > |12.8%|7.2%|2.6%|
> > > > >
> > > > >  As shown in the table below, and in line with [3], when using YaRN alone, the model struggles with long context recall, with its performance continuously dropping the longer the context. We hope these new results will help better highlight the critical difference of our method, and further support the rest of our results.
> > > > >
> > > > > > 2. Do you know if DroPE is better than YaRN because it has more training time during the recalibration phase? Can you compare it against YaRN but with the same recalibration phase?
> > > > >
> > > > > Since our recalibration phase is performed at the original context length, training YaRN with the same recipe is equivalent to continued pretraining of the base model. This is because the scaling factor of YaRN is directly linked to the context extension factor, resulting in an identity transformation when recalibrating at the same context. As demonstrated in our first set of experiments, in this setting DroPE matches that in-context performance of the base model (Figure 2), which is equivalent to the YaRN version, and outperforms YaRN on needle-in-a-haystack tasks (Table 1). Following the reviewer's feedback, we realized how this important property was never explicitly explained in our text, and we now extended Section 4 of the latest revision to make this clearer.
> > > > >
> > > > > > 3. Do you think that gradually removing the RoPE during the recalibration phase be better than outright removing it in a single step? It's highly likely that DroPE is equivalent to setting a RoPE Linear scaling to infinity.
> > > > >
> > > > > We thank the reviewer for this thought-provoking question. To test this, **we added an experiment to the latest revision of our work where the model's linear scaling is increased to $s=2$ (RoPE frequencies are multiplied by $0.5$) for the first half of recalibration before RoPE is removed entirely for the second half of recalibration**. We find that this, in fact _harms_ recalibration performance, leading to final perplexity which is higher by 11%.
> > > > >
> > > > > ||DroPE|Gradual DroPE|
> > > > > |-|-|-|
> > > > > |Final validation preplexity | **21.1** | 23.4 |
> > > > >
> > > > > One possible explanation is that gradually removing RoPE forces the model to constantly re-adjust itself to different distance compression rates instead of learning to extrapolate position from the causal mask alone.
> > > > >
> > > > > > 4. What happens with DroPE's performance at higher context scaling lengths? 4x, 8x, etc?
> > > > >
> > > > > See answer to weakness 2.
> > > > >
> > > > > ---
> > > > >
> > > > > **References**
> > > > >
> > > > > [1] An, Chenxin, et al. "Training-free long-context scaling of large language models." arXiv preprint arXiv:2402.17463 (2024).
> > > > >
> > > > > [2] Yang, An, et al. "Qwen3 technical report." arXiv preprint arXiv:2505.09388 (2025).
> > > > >
> > > > > [3] Lu, Yi, et al. "A controlled study on long context extension and generalization in llms." arXiv preprint arXiv:2409.12181 (2024).
> > > > >
> > > > > [4] Sahoo, Subham, et al. "Simple and effective masked diffusion language models." Advances in Neural Information Processing Systems 37 (2024): 130136-130184.
> > > > >
> > > > > [5] Nie, Shen, et al. "Large language diffusion models." arXiv preprint arXiv:2502.09992 (2025).
> > > > >
> > > > > [6] https://huggingface.co/datasets/HuggingFaceFW/fineweb-edu
> > > > >
> > > > > [7] Grattafiori, Aaron, et al. "The llama 3 herd of models." arXiv preprint arXiv:2407.21783 (2024).
> > > > >
> > > > > [8] Shang, Ning, et al. "LongRoPE2: Near-Lossless LLM Context Window Scaling." arXiv preprint arXiv:2502.20082 (2025).
> > > > >
> > > > > [9] Press, Ofir, Noah A. Smith, and Mike Lewis. "Train short, test long: Attention with linear biases enables input length extrapolation." arXiv preprint arXiv:2108.12409 (2021).
> > > > >
> > > > > [10] Vasylenko, Pavlo, et al. "Long-context generalization with sparse attention." arXiv preprint arXiv:2506.16640 (2025).
> > > > >
> > > > > [11] Allal, Loubna Ben, et al. "SmolLM2: When Smol Goes Big--Data-Centric Training of a Small Language Model." arXiv preprint arXiv:2502.02737 (2025).
> > > > >
> > > > > [12] Hsieh, Cheng-Ping, et al. "RULER: What's the Real Context Size of Your Long-Context Language Models?." arXiv preprint arXiv:2404.06654 (2024).

---

> ### Comment · Reviewer_iW8i · 2025-11-27
>
> Thank you for the extensive follow-up to my questions and concerns, **I am fairly satisfied with the results and will raise my score from a 4 to a 6**. The only thing that I am still doubtful about is that you claim that DroPE is free while YaRN is not, my original question was what would happen if you were to use YaRN with a large scaling factor of let's say s=32 and trained with the same recipe as DroPE? YaRN also exhibits zero shot context extension capabilities even if you keep your training data at the same context, the biggest contribution of the context extension comes from increasing the scaling factor (in short, most models using YaRN don't change the data mixture or increase the data's context length but simply increase the scaling factor *s* to a bigger value during continual pre-training) Only ABF/NTK scaling requires such a data mixture change.
>
> My main concern was: what happens if you set YaRN to s=32 (or some other big-ish value) and train as if you were training DroPE? It's as free as using DroPE while being a much fairer comparison.
> If the authors are willing to conduct a convincing and fair experiment I will increase the score even further. This paper's contributions might be significant to the community.

---

### Official Review · Reviewer_Zi8P · 2025-11-08

**Soundness:** 1
**Presentation:** 2
**Contribution:** 2
**Rating:** 2
**Confidence:** 4

**Summary:**

This paper introduces DroPE, a method that removes RoPE after pretraining to address its limitations. The authors demonstrate that while RoPE provides crucial inductive bias for rapid convergence during training, its explicit positional encoding hinders zero-shot generalization to longer sequences. They analyze why existing RoPE-scaling techniques fail in zero-shot scenarios by showing how they distort low-frequency attention heads essential for long-range semantic understanding. The core innovation is DroPE, which eliminates RoPE and employs a brief recalibration phase, compelling the model to rely on implicit positional cues from the causal mask and data patterns. In experiments, this approach achieves robust zero-shot extrapolation when extending the context length to twice the training length, significantly outperforming complex RoPE-scaling methods.

**Strengths:**

1. The paper shows that RoPE trained from scratch outperforms NoPE: NoPE has higher perplexity and exhibits gradient-vanishing issues during training.
2. The authors propose a new way to obtain a NoPE base model by “dropping” RoPE from a well-trained RoPE model (Drop RoPE).
3. They evaluate on selected datasets (NIAH and four LongBench subsets) and empirically demonstrate that RoPE→NoPE bases achieve length-generalization benefits for limited extrapolation ranges (mainly around 2×).

**Weaknesses:**

1. To me, the manuscript reads more like a blog post than a formal paper. The central claim—that DropRoPE provides a generalization advantage—feels weak across several experimental dimensions: the variety of base models tested, the extrapolation distances considered, the number and comprehensiveness of benchmarks, and the choice of baselines.
2. The observation that performance can be restored by continued training after dropping positional encodings has been reported elsewhere. For example, Table 5 in [1] shows smollm-135M can recover performance via continued training even when positional encodings are fully or partially removed.

[1] Towards Economical Inference: Enabling DeepSeek’s Multi-Head Latent Attention in Any Transformer-based LLMs (https://arxiv.org/pdf/2502.14837v1)

**Questions:**

1. Can NoPE extrapolation methods such as [2] be integrated with DropRoPE?
2. A 2× extrapolation advantage is likely of limited practical significance. Can you evaluate generalization over a wider range of lengths (for example, from 2× up to 8×, as in [2])?
3. Why were only four LongBench subsets selected rather than the full LongBench suite?
4. Can the method be validated on widely used LLMs (e.g., LLaMA, Qwen)?

[2] Length Generalization of Causal Transformers without Position Encoding (https://arxiv.org/pdf/2404.12224)

---

> ### Author Response · Authors · 2025-11-25
> **Authors' Response to Reviewer Zi8P (Part 1/4)**
>
> We thank the reviewer for their careful reading of our paper and for the constructive feedback; below, we clarify our contributions and address each of the raised comments and suggestions in turn.
>
> **Weaknesses**
>
> > 1. [...] The central claim—that DropRoPE provides a generalization advantage—feels weak across several experimental dimensions: the variety of base models tested, the extrapolation distances considered, the number and comprehensiveness of benchmarks, and the choice of baselines.
>
> To address this, we have now **added new experiments and an extended analysis** to Section 5 of the updated manuscript, and provide a larger **variety of base models** at even **larger scales**, more **comprehensive evaluation** with **longer extrapolation distances**, and **additional baselines**. We divide our response into three parts, which elaborate on each of these components and summarize the main analysis and results:
>
> **Part 1: Base models tested.** In Section 5 of our initial submission, we run experiments considering two main settings using different model scales:
> - We used a 500M parameter RoPE transformer trained to $2\times$ the chinchilla optimal rate to show that DroPE can be integrated as a *zero cost* extension at the end of standard pretraining pipelines. The goal of this setup is to perform targeted controlled experiments and compare DroPE with prior baselines.
> - We used the SmolLM-360M model pretrained on 600B tokens to show that DroPE can be applied to *adapt LLMs in the wild* at a fraction of their original training budget and study how far this can be pushed in terms of sample efficiency.
>
> Following the reviewer's feedback, we have added new experiments to the latest revision, considering **two additional pretrained LLMs** with increasing **numbers of parameters** and pretraining **context lengths**:
>
> First, we considered a larger 1.7B SmolLM model that was trained on 1T tokens. We applied DroPE and then recalibrated this model on only 20B tokens, 2% of the original training budget. As shown in the table below, despite the much lower recalibration budget in relative terms, our findings for this larger model are consistent with the ones obtained using the 360M variant, with DroPE outperforming other context extension techniques.
>
>   | Dataset       | Base  | NTK   | YaRN  |   PI  |   DroPE   |
>   |---------------|-------|-------|-------|-------|-----------|
>   | MultifieldQA  | 4.12  | 27.58 | 27.60 | 4.12  | **32.18** |
>   | MusiQue       | 0.50  | 3.37  | 3.90  | 0.50  | **7.53**  |
>   | GovReport     | 4.70  | 24.65 | 17.19 | 4.70  | **24.77** |
>
> Second, we considered Llama2-7B [2], originally pretrained on 4T tokens with 4096 context length, recalibrated once again for 20B tokens, which this time comprise only 0.5% of the original budget. We chose this model as the Llama family was suggested in the reviewer's question 4. We note that due to model scale and the increased context, merely adapting this model for 20B tokens already took more than 2000 H100 GPU hours. Furthermore, as the team behind Llama2-7B did not disclose the details of their pretraining data, we still resorted to the SmolLM training corpus. Nonetheless, despite having had to push the recalibration phase budget to the minimum in relative terms and not having access to the Llama closed-source corpus, we found our Llama7B-DroPE model still outperforms all RoPE-context extension scaling baselines on LongBench.
>
>   | Dataset      | Base  | NTK   | YaRN  | DroPE |
>   |-------------|-------|-------|-------|-------|
>   | MultifieldQA| 17.26 | 21.81 | 23.13 | **25.90** |
>   | MusiQue     | 10.43 | 10.91 | 7.65  | **12.88** |
>   | GovReport   | 32.41 | 32.91 | 26.65 | **39.47** |
>
> We hope these new results will significantly reinforce the evidence of the generality and scalability of DroPE, and that our efforts will help address the reviewer's feedback about the diversity and scale of base models.

---

> ### Author Response · Authors · 2025-11-25
> **Authors' Response to Reviewer Zi8P (Part 2/4)**
>
> **Part 2: Extrapolation distances considered and comprehensiveness of benchmarks.** Our original evaluation considered seven pretraining tasks [1] for in-context evaluation, three synthetic tasks from the RULER benchmark [3] where we chose to evaluate at 2x our model's training context length, and five challenging LongBench tasks with up to 32K tokens corresponding to 16x of our model's original context length. As suggested in the reviewer's feedback, we **extended our evaluation settings** with a detailed analysis of how varying the context extension lengths affects performance, and **evaluated our method on additional tasks and benchmarks**.
>
> First, to analyze how varying context length affects performance at a granular level beyond 2x, we started by evaluating our models on NIAH tasks of up to 8x of our original context length:
>
>   | Context length            |   DroPE   |  NTK  | YaRN  |
>   |---------------------------|-----------|-------|-------|
>   | 4K (2x original ctx)      | **74.92** | 29.84 | 48.25 |
>   | 8K (4x original ctx)      | **55.00** | 14.37 | 25.62 |
>   | 16K (8x original ctx)     | **52.50** |  7.19 | 12.18 |
>
> Additionally, we also recorded granular performances by considering the scores attained by the different context extension strategies, by partitioning LongBench problems by their input context lengths. To remove confounding factors, we focused this analysis on two tasks that include problems across all our considered context length evaluation brackets up to x8 and x16:
>
> **Multifield QA**
> | model        | 0-4K (0-2x ctx) | 4-8K (2-4x ctx) | 8-16K (4-8x ctx) |
> | ------------ | --------------- | --------------- | ---------------- |
> | SmolLM-DroPE | 32.82           | **24.73**       | **30.07**        |
> | SmolLM-NTK   | **34.25**       | 22.3            | 21.63            |
> | SmolLM-YaRN  | 33.96           | 22.91           | 20.08            |
>
> **MusiQue**
>
> | model        | 0-4K (0-2x ctx) | 4-8K (2-4x ctx) | 8-16K (4-8x ctx) | 16-32K (8-16x ctx) |
> | ------------ | --------------- | --------------- | ---------------- | ------------------ |
> | SmolLM-DroPE | **50.00**          | 6.11            | **8.05**         | **16.67**          |
> | SmolLM-NTK   | 0.0             | 4.36            | 3.36             | 0.0                |
> | SmolLM-YaRN  | 0.0             | **19.68**       | 3.13             | 7.14               |
>
> As shown by our new analysis results above, DroPE outperforms other RoPE-extension baselines across all extension factors. Interestingly, these results also reveal that beyond a 4x context extension factor, DroPE's performance appears to stabilize while the other methods sharply deteriorate. This is most evident on the synthetic NIAH tasks, where we are able to precisely control the distance to the information to retrieve for success. In this instance, we see DroPE's performance drops by only 2.5% (from 55% to 52.5%) when doubling the context length from 4x to 8x. In contrast, when applying NTK and YaRN, model performance halves (14.37% to 7.19% for NTK, and 25.62% to 12.18% for YaRN). We believe these results are in line with our motivating observations and theory: zero-shot context extension inherently conflicts with the presence of RoPE positional encodings (see Section 4), with DroPE being a practical method to break this paradigm without losing RoPE's substantial training benefits.
>
> Second, to expand the comprehensiveness of our evaluation, we started by expanding our LongBench evaluation to the full benchmark of 26 tasks. We note that in our original work, we chose to focus on five particular tasks as we wanted to minimize confounding factors due to the noisiness and difficulty of some tasks in the LongBench benchmark [4], where even powerful closed-source GPT models achieve less than 5% (e.g, Llama3 [5] also only evaluates on 2 tasks from the suite)
>
> | Dataset                      | Base | PI   | NTK   | YaRN  | DroPE |
> |------------------------------|------|------|-------|-------|-------|
> | Average (all LongBench tasks)| 2.59 | 2.48 | 12.21 | 13.07 | **13.81** |
>
>
> Moreover, we also considered a new long document evaluation setup, running our model on the full NarrativeQA [13] in its original settings without filtering. We evaluated using Llama2-7B as the base model and compared it to both YaRN and our DroPE version. Results are reported as ROUGE-L scores.
>
> |Base|YaRN|DroPE|
> |-|-|-|
> |0.0|6.98|**10.93**|
>
> In general, we find that DroPE excels at tasks that require reasoning or analysis on multiple pieces of data that appear deep in the context. These types of tasks require strong recall capabilities for which semantic entropy heads, that are preserved by DroPE, are particularly useful.

---

> ### Author Response · Authors · 2025-11-25
> **Authors' Response to Reviewer Zi8P (Part 3/4)**
>
> **Part 3: Choice of baselines.** In our original paper, we compared DroPE with zero-shot PI [6], YaRN [7], and NTK-aware [8] context extension, together with NoPE [9] and RNoPE-SWA [14] trained in our same controlled model budgets. Following the reviewer's feedback, **we have now added two additional baselines**, including the recent **LongRoPE2** [10] context extension and **AliBi** [11], which we trained from scratch using the same protocol as for our controlled experiments:
>
> **SmolLM post-training adaptation**
> | Context length            |   DroPE   |  NTK  | YaRN  | LongRoPE2 |
> |---------------------------|-----------|-------|-------|-----------|
> | 4K (2x original ctx)      | **74.92** | 29.84 | 48.25 |   44.20   |
> | 8K (4x original ctx)      | **55.00** | 14.37 | 25.62 |   26.20   |
> | 16K (8x original ctx)     | **52.50** |  7.19 | 12.18 |   16.45   |
>
> **Models trained from scratch**
> | Method                                  | Multi-Query | Multi-Key | Multi-Value |
> |-----------------------------------------|------------:|----------:|------------:|
> | RoPE transformer                        |        0.0  |      0.0  |        0.0  |
> | RoPE transformer + PI                   |        0.0  |      0.0  |        0.0  |
> | RoPE transformer + RoPE-NTK             |       21.1  |     19.4  |       16.5  |
> | RoPE transformer + YaRN                 |       17.8  |      0.5  |       14.6  |
> | RNoPE-SWA transformer                   |        5.2  |     25.6  |       20.6  |
> | NoPE transformer                          |        9.2  |     36.2  |       21.4  |
> | ALiBi transformer                       |        5.2  |      0.0  |        1.1  |
> | DroPE transformer                       |    **28.0**  |   **41.6**  |  **23.3**  |
>
> We note that ALiBi models have been known to struggle with needle-in-a-haystack tasks, and these results are consistent with past literature [12]. We posit that this is due to the persistent positional bias that, together with the attention softmax, would inevitably downweigh the semantic attention heads required for NIAH tasks.
>
> > 2. The observation that performance can be restored by continued training after dropping positional encodings has been reported elsewhere. For example, Table 5 in [1] shows smollm-135M can recover performance via continued training even when positional encodings are fully or partially removed.
>
> We thank the reviewer for referring us to this work, which focuses on finetuning general multi-head attention modules into MLA. This is related to our work, as it demonstrates another form of post-training architectural modification. **We have added a reference to this work and explained its connection to DroPE in Section 6**. Additionally, we note that the mentioned table appears in the appendix and is not reported as a major contribution or result of the paper, and that the results are not a full reconstruction of the in-context performance and show a gap of up to 9.14 in the performance of the NoPE model compared to the base model.
>
> **Questions**
>
> > 1. Can NoPE extrapolation methods such as [2] be integrated with DropRoPE?
>
> Yes, the method introduced in the mentioned paper is orthogonal to ours and shows how to achieve better length generalization given a NoPE-transformer LM, whereas we focus on how to get a high-performing NoPE-transformer in the first place. In fact, we use the first technique mentioned by the authors in our experiments, and report the optimal scaling factors in Appendix C. Following the reviewer's comment, **we have now modified the third paragraph of Section 5.1 to explicitly note this compatibility**.
>
> > 2. A 2× extrapolation advantage is likely of limited practical significance. Can you evaluate generalization over a wider range of lengths (for example, from 2× up to 8×, as in [2])?
>
> Following the reviewer's feedback, we added results on sequences up to 16x longer than the pretraining context (see part 2 of our response to weakness 1 for details).
>
> **NIAH**
> | Context length            |   DroPE   |  NTK  | YaRN  |
> |---------------------------|-----------|-------|-------|
> | 4K (2x original ctx)      | **74.92** | 29.84 | 48.25 |
> | 8K (4x original ctx)      | **55.00** | 14.37 | 25.62 |
> | 16K (8x original ctx)    | **52.50** |   7.19 | 12.18 |
>
> **Multifield QA**
> | model        | 0-4K (0-2x ctx) | 4-8K (2-4x ctx) | 8-16K (4-8x ctx) |
> | ------------ | --------------- | --------------- | ---------------- |
> | SmolLM-DroPE | 32.82           | **24.73**       | **30.07**        |
> | SmolLM-NTK   | **34.25**       | 22.3            | 21.63            |
> | SmolLM-YaRN  | 33.96           | 22.91           | 20.08            |
>
> **MusiQue**
> | model        | 0-4K (0-2x ctx) | 4-8K (2-4x ctx) | 8-16K (4-8x ctx) | 16-32K (8-16x ctx) |
> | ------------ | --------------- | --------------- | ---------------- | ------------------ |
> | SmolLM-DroPE | **50.0** | 6.11 | **8.05** | **16.67** |
> | SmolLM-NTK   |  0.0| 4.36 | 3.36 | 0.0 |
> | SmolLM-YaRN  | 0.0 |**19.68**| 3.13| 7.14|

---

> > ### Author Response · Authors · 2025-11-25
> > **Authors' Response to Reviewer Zi8P (Part 4/4)**
> >
> > > 3. Why were only four LongBench subsets selected rather than the full LongBench suite?
> >
> > As suggested by the reviewer, we now consider additional challenging evaluation settings, including the full range of LongBench tasks, with extended context lengths up to 16x, showing that DroPE remains effective, outperforming highly popular RoPE-scaling baselines (see part 2 of our response to weakness 1 for details).
> >
> > > 4. Can the method be validated on widely used LLMs (e.g., LLaMA, Qwen)?
> >
> > As suggested by the reviewer, we now extended a Llama2-7B in large-scale experiments, validating that our methodology can be applied at only 0.5% of the original model's training budget, and be effective even in the presence of data mismatch, as the team behind the Llama paper did not disclose the details of the pretraining corpus (see part 1 of our response to Weakness 1 for details).
> >
> > ---
> >
> > **References**
> >
> > [1] Allal, Loubna Ben, et al. "SmolLM2: When Smol Goes Big--Data-Centric Training of a Small Language Model." arXiv preprint arXiv:2502.02737 (2025).
> >
> > [2] Touvron, Hugo, et al. "Llama 2: Open foundation and fine-tuned chat models." arXiv preprint arXiv:2307.09288 (2023).
> >
> > [3] Hsieh, Cheng-Ping, et al. "RULER: What's the Real Context Size of Your Long-Context Language Models?." arXiv preprint arXiv:2404.06654 (2024).
> >
> > [4] Bai, Yushi, et al. "Longbench: A bilingual, multitask benchmark for long context understanding." Proceedings of the 62nd Annual Meeting of the Association for Computational Linguistics (Volume 1: Long Papers). 2024.
> >
> > [5] Grattafiori, Aaron, et al. "The llama 3 herd of models." arXiv preprint arXiv:2407.21783 (2024).
> >
> > [6] Chen, Shouyuan, et al. "Extending context window of large language models via positional interpolation." arXiv preprint arXiv:2306.15595 (2023).
> >
> > [7] Peng, Bowen, et al. "Yarn: Efficient context window extension of large language models." arXiv preprint arXiv:2309.00071 (2023).
> >
> > [8] emozilla. Dynamically Scaled RoPE further increases performance of long context LLaMA with zero fine-tuning, 2023. URL https://www.reddit.com/r/LocalLLaMA/comments/
> >
> > [9] Kazemnejad, Amirhossein, et al. "The impact of positional encoding on length generalization in transformers." Advances in Neural Information Processing Systems 36 (2023): 24892-24928.
> >
> > [10] Shang, Ning, et al. "LongRoPE2: Near-Lossless LLM Context Window Scaling." arXiv preprint arXiv:2502.20082 (2025).
> >
> > [11] Press, Ofir, Noah A. Smith, and Mike Lewis. "Train short, test long: Attention with linear biases enables input length extrapolation." arXiv preprint arXiv:2108.12409 (2021).
> >
> > [12] Vasylenko, Pavlo, et al. "Long-context generalization with sparse attention." arXiv preprint arXiv:2506.16640 (2025).
> >
> > [13] Kočiský, Tomáš, et al. "The narrativeqa reading comprehension challenge." Transactions of the Association for Computational Linguistics 6 (2018): 317-328.
> >
> > [14] Yang, Bowen, et al. "Rope to nope and back again: A new hybrid attention strategy." arXiv preprint arXiv:2501.18795 (2025).

---

> > > ### Comment · Reviewer_Zi8P · 2025-11-28
> > >
> > > Thank you for the detailed response. I have no further questions.
> > >
> > > The paper's core contribution identifies an alternative approach for obtaining an effective NoPE backbone. This claim is now well-supported by the extensive experiments presented in the rebuttal. I will raise my score from 2 to **6**. If the initial submission included all or some of these additional experiments, I would have been more inclined to positively support this paper.

---

### Author Response · Authors · 2025-12-03
**Author's Final Remarks for Reassigned AC**

Among reviewers, there was a general appreciation of our methods' novelty, theoretical contribution, and potential impact, e.g., reviewer **iW8i** wrote:

> Novel and counterintuitive hypothesis where positional embeddings might not be necessary throughout a model's lifecycle.

> Treating PEs (Positional Embeddings) as "training scaffolds" that can be removed is elegant and very good for downstream use cases such as long context fine-tuning and kv-cached inference performance.

> Good theoretical contributions on why PEs are necessary during training and why they can be removed after.

> Strong potential for practical impact because of how simple this method is. Furthermore, reduces the inductive biases of LLMs after removing RoPE.

The reviewers' concerns were constrained to three shared axes:

1. Model scale: In our original version, we applied our method to (i) a 500M parameters LM we trained from scratch (for mor than $2\times$ chinchilla optimal number of tokens) to show that our method can be used as a _zero-cost_ extension at the end of standard pretraining, and (ii) a pretrained SmolLM-360M model that was originally pretrained on 600B tokens, to show that our method can be used to adapt LLMs in the wild at a fraction of their original training budget. Reviewers asked for experiments with a larger model, mentioning the Llama model family as a target option.


    - During the discussion period, **we added results showing that our method can successfully extend the context of the larger SmolLM-1.7B model, originally pretrained on 1T tokens, and of the Llama2-7B model, originally pretrained on 4T tokens**. Recalibrating Llama2-7B with our method only required 20B tokens at the pretraining context (0.5% of the original training cost), yielding marked improvements across all considered long-context settings, outperforming all baselines, without any long context training.

2. Extrapolation length: In the original version of the paper, we evaluated our models on tasks from the RULER benchmark at $2\times$ the original context, and on 4 tasks from the Longbench benchmark, which contain instances of over 32K tokens (16x the original context). Unfortunately, reviewers **Zi8P**, **iW8i**, and **PvVC** seemed to have missed our long context evaluation and have mistakenly thought that we had only validated our method on $2\times$ the original context.


    - During the discussion period, we clarified this important point in our work and in the responses, providing a breakdown of the context length of our evaluation tasks. We also added new results, evaluating our model on up to 8x its context size on RULER, and added more evaluation domains, including the entire LongBench and NarrativeQA suites.

3. Baselines and ablations: In the original version, we compare against three RoPE-scaling methods as well as two specialized architectures designed for length generalization. We performed ablations with respect to the recalibration cost. Several reviewers asked us to consider additional baselines, mentioning LongRoPE2, and perform ablations with respect to the recallibration starting point.


    - During the discussion period, **we have added comparisons against two new baselines**: the LongRoPE2 extension method (integrated on top of SmolLM) and the ALiBi positional embedding scheme (which we pretrained from scratch on top of the 500M parameter model with the same hyperparameters). We also added the requested ablations, which further reinforce our validation for the theoretical analysis.

**Paper status before reassignment**

During the discussion period and before the paper reassignment, three out of four reviewers (**Zi8P**, **iW8i**, and **fTZx**) have had the chance to read our clarifications and reply to our comments. As documented in the discussion below, **all of these three have explicitly stated their intention to increase their scores in favor of acceptance**: Reviewer **Zi8P** wrote they would increase their score from 2 to 6, reviewer **iW8i** from 4 to 6, and reviewer **fTZx** from 6 to 7. This leaves our paper with a majority positive assessment, with the final reviewer (**PvVC**) not having time to acknowledge our comments.

While we believe reviewer **PvVC**'s concerns are very close to all other reviewers, we understand that, given the current situation, they will not have the opportunity to respond. Despite this, we still hope that our rebuttal discussion and clarifications will have a chance to be reviewed and will weigh in on the final judgment of this work.

Regardless, we respect the conference's decision and sincerely appreciate the invaluable work of the ACs in this unprecedented situation.

The Authors

---

### Meta-Review · Area_Chair_uw9E · 2025-12-08

**Summary:**

This paper introduces DroPE, a simple yet counterintuitive method for long-context generalization in transformers that treats positional embeddings (specifically RoPE) as training-time scaffolds rather than permanent architectural components. The key idea is to pretrain with RoPE for efficient optimization, then remove positional embeddings entirely and perform a short recalibration phase at the original context length, yielding an effective NoPE model with strong zero-shot extrapolation. The paper combines theoretical analysis explaining (i) why NoPE models are difficult to train from scratch and (ii) why RoPE-scaling methods fundamentally struggle with zero-shot long-context retrieval, with extensive empirical validation across model scales and benchmarks.

Across the four reviews, the paper elicited strong interest for its novelty, simplicity, and potential impact, but initially faced skepticism regarding scale, extrapolation length, baselines, and experimental rigor. Crucially, after the rebuttal and added experiments, three of the four reviewers explicitly updated their scores upward and expressed support for acceptance. The remaining reviewer did not update their score, but their concerns substantially overlapped with those raised by the others and were directly addressed in the revised experiments and analysis.

**Reviewer Concerns:**

Addressed:
* Model scale and realism were strengthened with added experiments on larger models such as SmolLM-1.7B and Llama2-7B, showing effectiveness with very small recalibration budgets.
* Extrapolation range was clarified and extended with evaluations up to 8-16X context length on NIAH, the full LongBench suite, and NarrativeQA.
* Baselines were expanded to include stronger and more recent methods such as LongRoPE2 and ALiBi, alongside extended comparisons to NTK, YaRN, PI, and NoPE variants.
* Experimental rigor was improved through detailed ablations on recalibration cost, timing of RoPE removal, training stability, and architectural details such as QKNorm, with clearer experimental controls.
* Comparisons to YaRN were clarified by distinguishing zero-shot extension from approaches that rely on expensive long-context mid-training, supported by new zero-shot experiments.

Partially unresolved:
* Extremely large extrapolation factors such as >64k tokens are not evaluated in a fully zero-shot setting.
* Some theoretical components remain primarily explanatory rather than prescriptive for selecting optimal hyperparameters or recalibration schedules.

**Reviewer Scores:**

* Reviewer Zi8P (2): explicitly updated to 6.
* Reviewer iW8i (4): explicitly updated to 6.
* Reviewer fTZx (6): explicitly updated to 7.
* Reviewer PvVC (2): likely +2.

---

### Decision · Program_Chairs · 2026-01-26

Accept (Poster)